# PP2A/B55α substrate recruitment as defined by the retinoblastoma-related protein p107

Holly Fowle[1], Ziran Zhao[1], Qifang Xu[2], Jason S Wasserman[1], Xinru Wang[3], Mary Adeyemi[1], Felicity Feiser[1], Alison N Kurimchak[1], Diba Atar[1], Brennan C McEwan[4], Arminja N Kettenbach[4], Rebecca Page[5], Wolfgang Peti[6], Roland L Dunbrack[2], Xavier Graña[1]*

[1]Fels Cancer Institute for Personalized Medicine, Temple University Lewis Katz School of Medicine, Philadelphia, United States; [2]Institute for Cancer Research, Fox Chase Cancer Center, Philadelphia, United States; [3]Department of Chemistry and Biochemistry, University of Arizona, Tucson, United States; [4]Department of Biochemistry and Cell Biology, Hitchcock Medical Center at Dartmouth, Lebanon, United States; [5]Department of Cell Biology, UConn Health, Farmington, United States; [6]Department of Molecular Biology and Biophysics, UConn Health, Farmington, United States

*For correspondence:
xgrana@temple.edu

Competing interest: The authors declare that no competing interests exist.

**Abstract** Protein phosphorylation is a reversible post-translation modification essential in cell signaling. This study addresses a long-standing question as to how the most abundant serine/threonine protein phosphatase 2 (PP2A) holoenzyme, PP2A/B55α, specifically recognizes substrates and presents them to the enzyme active site. Here, we show how the PP2A regulatory subunit B55α recruits p107, a pRB-related tumor suppressor and B55α substrate. Using molecular and cellular approaches, we identified a conserved region 1 (R1, residues 615–626) encompassing the strongest p107 binding site. This enabled us to identify an 'HxRVxxV$_{619-625}$' short linear motif (SLiM) in p107 as necessary for B55α binding and dephosphorylation of the proximal pSer-615 in vitro and in cells. Numerous B55α/PP2A substrates, including TAU, contain a related SLiM C-terminal from a proximal phosphosite, 'p[ST]-P-x(4,10)-[RK]-V-x-x-[VI]-R.' Mutation of conserved SLiM residues in TAU dramatically inhibits dephosphorylation by PP2A/B55α, validating its generality. A data-guided computational model details the interaction of residues from the conserved p107 SLiM, the B55α groove, and phosphosite presentation. Altogether, these data provide key insights into PP2A/B55α's mechanisms of substrate recruitment and active site engagement, and also facilitate identification and validation of new substrates, a key step towards understanding PP2A/B55α's role in multiple cellular processes.

## Introduction

Protein phosphorylation is a reversible post-translational modification that is critical for the regulation of signaling and other cellular processes. It is estimated that a third of all cellular proteins are phosphorylated (*Ficarro et al., 2002*), with more than 98% of those phosphorylation events occurring on serine and threonine residues (*Olsen et al., 2006*). The opposing processes of phosphorylation and dephosphorylation are catalyzed by protein kinases and phosphatases, respectively. Despite the fundamental importance of dephosphorylation for normal cell physiology, the mechanisms of substrate recognition by protein phosphatases are only poorly understood (reviewed in *Brautigan and Shenolikar, 2018*).

Members of the *phosphoprotein phosphatase* (PPP) family of serine/threonine phosphatases are responsible for the majority of dephosphorylation in eukaryotic cells, with protein phosphatase 1 (PP1) and protein phosphatase 2A (PP2A) accounting for more than 90% of the total phosphatase activity (*Moorhead et al., 2007*; *Virshup and Shenolikar, 2009*). Within the PPP family, the molecular details of substrate recognition have been best studied for PP1 and calcineurin (or PP2B), and involve the recognition of defined short linear motifs (SLiMs) including RVxF, $\Phi\Phi$ (where $\Phi$ refers to a hydrophobic residue), SILK, among others for PP1 (*Choy et al., 2014*; *Kumar et al., 2018*); and LxVP and PxIxIT for PP2B. These motifs are characterized by the presence of three or four core interacting amino acids that are part of a 4–10 amino acid stretch within intrinsically disordered regions (IDRs) of regulator and/or substrate proteins. This SLiM-mediated specific targeting of substrates or regulatory proteins to PPPs is essential for the temporal and spatial coordination of its functions (reviewed in *Brautigan and Shenolikar, 2018*; *Heroes et al., 2013*).

PP2A is a highly conserved Ser/Thr protein phosphatase that makes up close to 1% of total cellular protein in some tissue types, making it one of the most abundant enzymes (*Fowle et al., 2019*). As a multimeric protein, PP2A can exist as a heterodimeric 'core enzyme,' consisting of a scaffold (A) subunit and a catalytic (C) subunit, or a heterotrimeric holoenzyme, in which the core dimer complexes with a B subunit (reviewed in *Shi, 2009*). The B subunit of PP2A can be subdivided into one of four major families (B55, B56, B72, and B93), each consisting of 2–5 isoforms and numerous splice variants. Apart from its role in subcellular localization, the B subunits are thought to be the key determinant of substrate specificity for the PP2A complex (reviewed in *Fowle et al., 2019*; *Kurimchak and Graña, 2012*; *Virshup and Shenolikar, 2009*). Indeed, a PP2A substrate SLiM (LxxIxE) has recently been identified for the B56 family of PP2A regulatory subunits, which form a HEAT repeat fold. The binding affinity of LxxIxE can be modulated by phosphorylation (phosphorylation leads to tighter binding), and it binds in a binding pocket between HEAT repeats 3 and 4. Furthermore, it was recently shown that additional factors including conserved, dynamic charged:charged interaction modulate substrate specificity for B56 (*Wang et al., 2020*). This combined molecular and cellular data can be leveraged to validate known B56/PP2A substrates and, most importantly, identify potential novel B56 substrates (*Hertz et al., 2016*; *Wang et al., 2016*). In contrast, most of this information is missing for all other B family, limiting our ability to understand substrate recruitment.

B55α (four isoforms, α, β, γ, δ; our work focuses on the α-isoform of B55) is ubiquitously expressed and the most abundant regulatory subunit of PP2A (*Kim et al., 2014*; *Wang et al., 2015*). Critically, all known PP2A/B55α-dependent substrates have key functions in cell division, differentiation, and survival, and are found to be dysregulated in cancer and Alzheimer's disease (reviewed in *Fowle et al., 2019*). The structure of the PP2A/B55α holoenzyme allowed for the suggestion of a mechanism for targeting the substrate Tau. Specifically, it was proposed the large negatively charged patch on B55α recruits Lys-rich domains of TAU (*Xu et al., 2008*), only to be shown that other substrates must use different molecular recognition motifs (*Jayadeva et al., 2010*). Recently, proteomics data suggested that B55α preferred substrates that are phosphorylated by Ser/Thr-Pro directed kinases (*Cundell et al., 2016*; *Zhao et al., 2019*). However, no substrate recruitment mechanism has been identified for B55α, and thus, further insights are of key importance to understand how substrates engage B55.

Here, by combining and leveraging molecular and cellular data, we characterize the recruitment of the PP2A/B55α-specific substrate p107 (*Garriga et al., 2004*; *Jayadeva et al., 2010*; *Kolupaeva et al., 2013*; *Kurimchak et al., 2013*). p107 is a pRB-related tumor suppressor with key regulatory roles in the cell cycle (reviewed in *Kurimchak and Graña, 2015*). Our data lead to the identification of a conserved SLiM in key B55α substrates and enable us to develop a structural model for B55α substrate recognition that enables us to predict how other substrates bind B55α. Since modulation of PP2A activity is actively being explored in many cancer types, our data will further help in turning PP2A into a key cancer drug target.

## Results
### p107 binds to B55
p107 (retinoblastoma-like protein 1 [RBL1]) is a multidomain protein. We have previously shown that the p107 'spacer' region (p107 residues 585–780, which link the retinoblastoma conserved regions A

and B) is sufficient for PP2A/B55α binding (*Jayadeva et al., 2010*). IUPred2a bioinformatics analysis (*Mészáros et al., 2018*) shows that this linker region is intrinsically disordered (*Figure 1A*). Aligning the intrinsically disordered p107 linker region from 60 different species enabled the identification of three highly conserved regions (termed R1, R2, and R3) (*Figure 1*, *Figure 1—figure supplement 1*). Indeed, the conservation of R1 and R2 extended to a conserved family member p130 (RBL2) (*Figure 1C*). Within the spacer, there are also three potential CDK kinase phosphorylation sites (p107 S615, S640, and S650).

## R1 is necessary and sufficient for B55α binding

To determine the contribution of each conserved region, we generated p107 deletion mutants and performed GST pull-down assays using U-2 OS whole-cell lysates. *Figure 1D* shows that a mutant lacking residues C-terminal of R2 binds B55α similarly to the full construct, indicating that residues C-terminal to the R2 domain are dispensable for B55α binding. R1 alone can bind B55α, although to a slightly lesser extent than R1/R2 (lane 3), while mutants lacking regions containing R1 do not bind B55α (lanes 6 and 7). Additional experiments highlighted that R1 is the key B55α binding region and that R2 enhances the interaction, but without R1 cannot recruit B55α alone (*Figure 1D*). Conversely, binding of p107 to CDK2 (or cyclin A, not shown) strictly depends on the presence of R2 (lanes 2, 4, 5, and 6), which includes an RxL motif necessary for cyclin A/CDK2 binding (*Adams et al., 1996*; *Chen et al., 1996*).

It has been shown that charged:charged interactions are central for B55α:TAU substrate recruitment. To test if charged:charged interactions are also important for p107 recruitment, we generated p107 R621A and K623A variants (in R1), R633A and R34A variants (in s2), and an R647A variant (in R2) using GST-p107-R1R2. As shown in *Figure 1D*, positive residues in R1 and R2, but not the connecting s2 'linker,' led to a reduction in binding to B55α. Mutation of the cyclin A binding site in R2 (659KRRL-AAAA and 660RRL-AAA mutations to Ala), which also contains positive residues, also reduced binding (data not shown). Thus, positively charged residues participate in binding to B55α.

We also used NMR spectroscopy to gain further atomic resolution insights into the interaction between p107 and B55α. The 2D [$^1$H,$^{15}$N] HSQC spectrum of $^{15}$N-labeled p107 (residues M612-S687, which include R1, R2, and R3) shows all hallmarks of an intrinsically disordered protein (IDP), with a highly limited proton chemical shift dispersion due to the lack of a hydrogen bond network in secondary structure elements. This experimentally validates the bioinformatics data. An overlay of the 2D [$^1$H,$^{15}$N] HSQC spectrum of $^{15}$N-labeled p107$_{M612-S687}$ in the presence and absence of purified B55α shows that ~15 peaks have reduced intensity, typical for an IDP:protein interaction (*Figure 1E and F*). Upon completion of the sequence-specific backbone assignment of p107$_{M612-S687}$, we identified that cross-peaks corresponding to p107 residues M614-E624 were most significantly broadened upon binding to B55α (*Figure 1F*). Consistent with the mutation data, these residues form the core of R1. Furthermore, residues within the linker region and R2 also showed peak intensity attenuations, consistent with additional weaker interactions in R2 contributing and enhancing the interaction of R1. Taken together, the NMR data reinforce that the p107 R1 region mediates key contacts with B55α. p107 interacts with specific B55α surface residues.

The recently identified LxxIxE B56 SLiM binds to a highly conserved binding pocket on B56. Thus, to identify B55α residues that mediate p107 binding, we analyzed B55α conservation. B55α adopts a β-propeller fold. The most highly conserved residues cluster at the top of the β-propeller and a fraction of them form a highly negatively charged patch (*Figure 2A and B*, *Figure 2—figure supplement 1*). Based on this analysis and the observation of a prominent groove on the surface of B55α blade 4 (*Figure 2A and B*, *Figure 2—figure supplement 1*) that might function as a potential guide for substrates to reach the catalytic site of PP2A/C, we generated 19 Myc-tagged B55α variants (either Ala substitutions or charge reversals; *Figure 2C*, see *Figure 2—figure supplement 1* for a complete list). All these residues except E27, which lies in a protein segment not present in B55γ, are conserved in the other three members of the B55 family. All 19 B55α variants were tested in co-immunoprecipitation assays to measure binding to p107. Furthermore, we also tested binding for two additional PP2A:B55α substrates, pRB and KSR1, to test if this B55α interaction surface is shared by different substrates.

For each substrate of interest, 293T cells were co-transfected with a FLAG-tagged substrate (p107, pRB, or KSR1) and wildtype or mutant Myc-B55α constructs. Representative assays and the

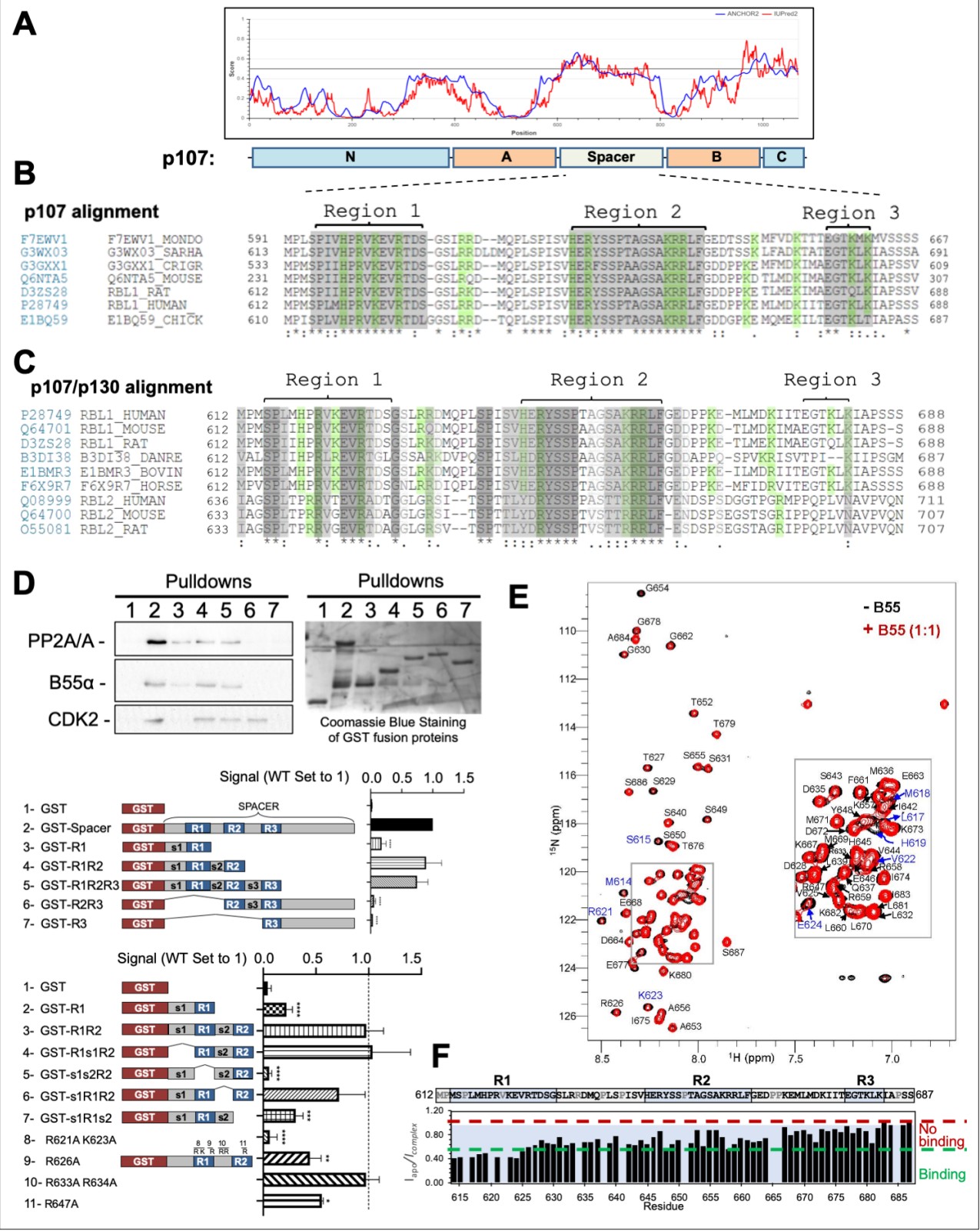

**Figure 1.** The intrinsically disordered spacer region of p107 contains three highly conserved regions (R1, R2, and R3), of which R1 is required for B55α binding and R2 enhances the binding interaction as determined via mutational analysis and NMR. (**A**) The spacer and the C-terminus of p107 are intrinsically disordered (IUPred2a web interface, *Sievers et al., 2011*). (**B**) Clustal W alignment of conserved amino acid sequences of the p107 spacer from different species. Three highly conserved regions within the spacer were identified, which are highlighted in gray and named as region 1 (R1),

*Figure 1 continued on next page*

*Figure 1 continued*

region 2 (R2), and region 3 (R3). Positively charged residues are highlighted in green. (**C**) Clustal W alignment of conserved amino acid sequences of the p107 and p130 spacer from the indicated species. Conserved residues are highlighted in shades of gray. Positively charged residues are highlighted in green. (**D**) A GST-p107 spacer construct was used as a template to systematically delete indicated regions and mutate positively charged amino acids in R1 and R2. The p107 spacer spans amino acids 584–782 and the small spacers, s1 (584–615), s2 (631–644), and s3 (662–676), separate the beginning of the spacer from R1 and the three conserved regions R1, R2, and R3. Pull-down assays with the indicated fusion proteins from U2-OS lysates were performed, and binding of the indicated proteins was determined by western blot analysis. GST-p107 was determined by Coomassie Blue or Ponceau S staining. Experiments were performed in triplicate, and quantification values represent the mean B55α/p107 variant ratios ± standard deviation (SD). (**E**) Overlay of the 2D [$^1$H, $^{15}$N] HSQC spectra of $^{15}$N-labeled p107 (M612-S687) in the presence (red) and absence (black) of purified monomeric full-length B55α (M1–N447) (see Materials and methods for B55α purification). (**F**) p107 (M612-S687) sequence is shown above the peak intensity ratios for data shown in (**E**). Prolines, residues not assigned (M612 and R634), and overlapping residue (V622) are labeled in grey. Residues corresponding to R1, R2, and R3 are highlighted in blue. Approximate binding cutoffs are marked by dashed lines.

The online version of this article includes the following figure supplement(s) for figure 1:

**Source data 1.** Uncropped replicates of western blot images and PVDF membranes stained with Coomassie Blue used to quantitate B55α binding to conserved regions in the p107 spacer (top bar graph).

**Source data 2.** Uncropped replicates of western blot images and PVDF membranes stained with Coomassie Blue or Ponceau S used to quantitate binding to B55α of conserved regions in the p107 R1–R2 construct (top bar graph) and R/K point mutant variants of p107 R1–R2 (lower bar graph).

**Figure supplement 1.** Clustal W alignment of conserved amino acid sequences of the p107 spacer from different species.

associated quantitation from the co-immunoprecipitation experiments are shown in *Figure 2D and E* and *Figure 2—figure supplement 1*. B55α D197K and L225A variants showed the most profound impact on the recruitment of all three substrates when directly compared to wildtype B55α. Other B55α mutants showed varying effects on binding depending on the substrate, as illustrated by B55α V228A affecting only p107 binding while B55α H179A affected p107 and KSR1 but not pRB binding (*Figure 2—figure supplement 1*). To ensure that B55α variants did not induce a change in overall protein conformation, we assessed binding of all B55α mutants to PP2A/C. B55α D197K was the only variant that had a minor effect on PP2A/A and PP2A/C recruitment (*Figure 2D*, left panel).

Taken together, while all substrates use the highly conserved top and groove of B55α for their interaction, our data demonstrate the contribution of distinct residues on the B55α surface for binding to different substrates (*Figure 2F*). The two B55α residues required by the three substrates tested, D197K and L225A, are located on the B55α blade 4, which directly points towards the active site of PP2A/C. All other residues are an extension of this core binding interaction between the substrates and B55α. p107 S615 and S640 are PP2A/B55α dephosphorylation sites.

PP2A/B55α is known to counteract phosphorylation by CDK kinases during mitosis. p107$_{M612-S687}$ contains three SP sites (S615, S640, and S650, *Figure 1C*) that can be phosphorylated in cells (*Hornbeck et al., 2015*). Using recombinant cyclin A/CDK2 and γ-$^{32}$P-ATP, we observed robust in vitro phosphorylation of p107. Our data suggested that at least two different sites are phosphorylated (*Figure 3—figure supplement 1*, top). We confirmed these phosphorylation sites using antibodies raised against the phospho-CDK substrate motif 'K/HpSP' (*Figure 3—figure supplement 1*, bottom).

To further analyze PP2A/B55α-mediated dephosphorylation, we purified recombinant PP2A/B55α (*Figure 3A*, right panel; *Zhao et al., 2019*) and used this functional holoenzyme to analyze time-dependent dephosphorylation of p107$_{M612-S687}$ (*Figure 3A*, left panel), which is inhibited by 10 nM of the PP2A/C-specific inhibitor okadaic acid. To determine which sites on p107 were dephosphorylated, we generated p107 mutants in which only a single 'SP' site can be phosphorylated (i.e., S615A-S640A, S615A-S650A, and S640A-S650A), as well as a triple deletion control (S615A-S640A-S650A) (*Figure 3B*). To validate phosphorylation/dephosphorylation and determine which sites were targeted by PP2A/B55α, we used mass spectrometry (MS). MS identified pS615 and pS640 in CDK2-phosphorylated p107 and a significant reduction of pS615 and pS640 phosphorylation upon incubation with PP2A/B55α (*Figure 3C*). Interestingly, S650 phosphorylation was not identified, likely due to the close proximity of S650 to the RXL$_{658-660}$ motif that mediates cyclin A/CDK2 binding to p107.

Next, we performed in vitro dephosphorylation experiments using p107 variants that are deficient in B55α binding due to elimination of positive charges in R1 or R1–R2. As expected, reduced dephosphorylation of p107 was detected for the binding-deficient p107 R1 [R621A/K623A] variant when compared to wt-p107, confirming our newly identified substrate binding residues (*Figure 4A*).

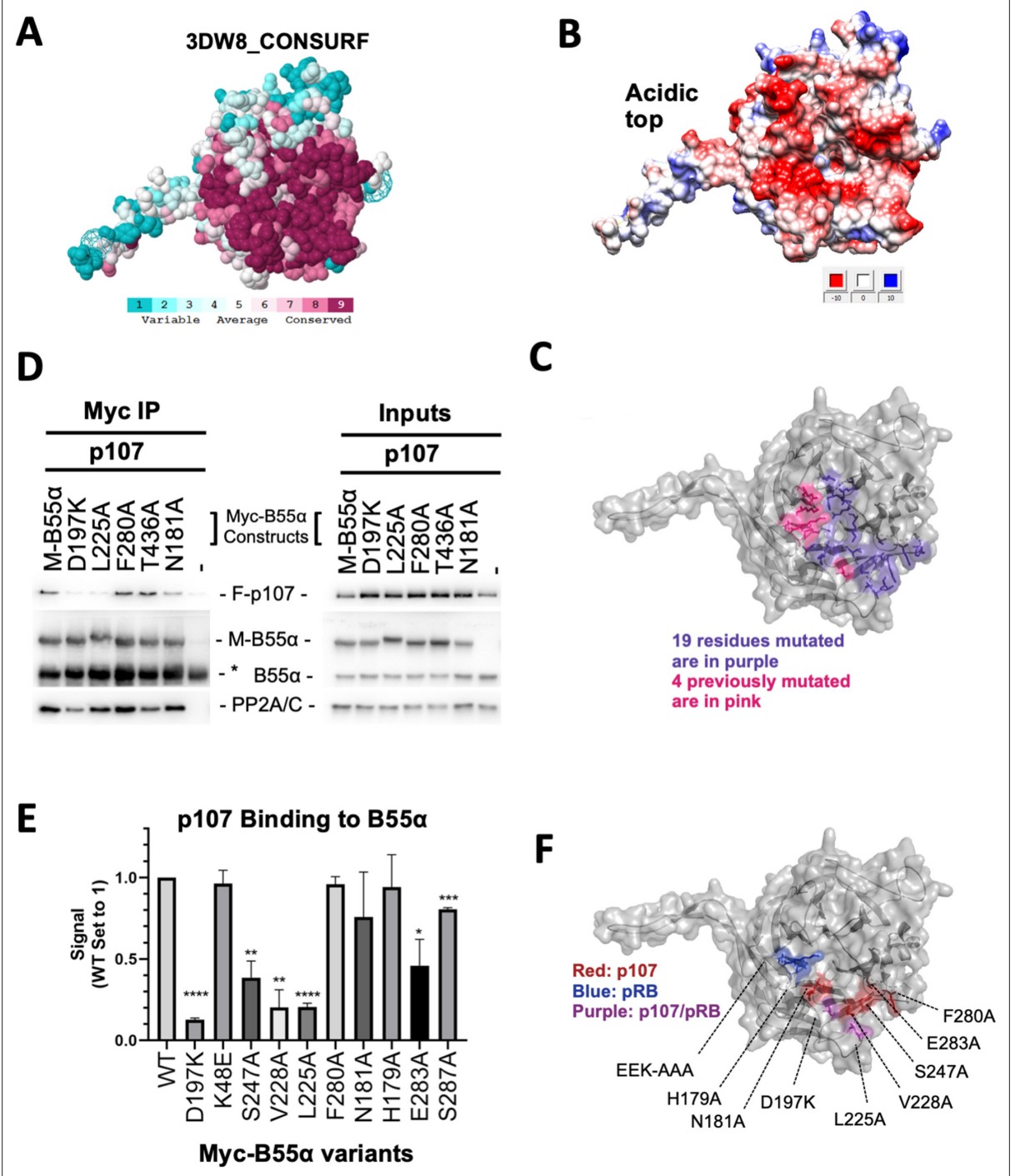

**Figure 2.** Mutation of highly conserved residues on the β-propeller top of B55α have substrate-specific effects on binding, supporting the notion that substrates contact different surfaces on B55α. (**A**) ConSurf depiction of PP2A/B55α mapping amino acid conservation (where amino acids are color-coded by conservation). (**B**) Electrostatic predictions mapped to the surface of the PP2A/B55α structure indicate an acidic top (red, acidic; blue, basic). (**C**) Nineteen single-point mutations on the conserved top of the B55α β-propeller were generated. These are shown in purple. Four mutations generated and analyzed previously are shown in pink. (**D**) Representative immunoprecipitation experiment to test p107 binding requirements on Myc-B55α. Flag-p107 and wild-type and mutant Myc-B55α mutant constructs were co-transfected into 293T cells and used for IPs with anti-Myc agarose conjugate. These assays were resolved via SDS-PAGE, and proteins were detected using anti-Flag, anti-B55α, and anti-PP2A/C. (**E**) Mean values for cumulative immunoprecipitation assays for Flag-tagged p107 binding to Myc-B55α constructs are shown, with statistics indicated above. Experiments were performed in triplicate or duplicate, and quantification values represent the mean p107:Myc-B55α variant ratios ± standard error. (**F**) The surface

*Figure 2 continued on next page*

Figure 2 continued

structure of B55α is depicted, with residues that appear important for p107 and pRB binding color-coded (in red and blue, respectively). The two Myc-B55α mutants that affect binding of both p107 and pRB (D197K and L225A) are colored purple.

The online version of this article includes the following figure supplement(s) for figure 2:

Source data 1. Upper, middle, and lower western blot membranes for *Figure 2D* (replicate 1).

Source data 2. Western blot membranes for replicates 1–3.

Source data 3. Western blot membranes for replicates 1–2 used for the quantitation shown in *Figure 2E*.

Figure supplement 1. Effect of B55α mutations on p107, pRB and KSR1 binding.

Figure supplement 1—source data 1. Western blot membranes for replicates used for the quantitation of Flag-pRB:Myc-B55α binding ratios shown in *Figure 2—figure supplement 1* (middle).

Figure supplement 1—source data 2. Western blot membranes for replicates used for the quantitation of Flag-pRB:Myc-B55α binding ratios shown in *Figure 2—figure supplement 1* (middle).

Figure supplement 1—source data 3. Western blot membranes for replicates used for the quantitation of Flag-pRB:Myc-B55α binding ratios shown in *Figure 2—figure supplement 1* (middle).

Figure supplement 1—source data 4. Upper, middle, and lower western blot membranes for *Figure 2—figure supplement 1*, bottom.

Dephosphorylation was further reduced for the p107-R1-R2 variant (R1 [R621A/K623A]-R2 [K657A/R659A]) (*Figure 4A*), correlating well with the NMR-based binding affinities.

Lastly, we also assessed PP2A/B55α-mediated dephosphorylation of substrates using competitive peptides, specifically a peptide encompassing the R1 region of p107. As expected, dephosphorylation of p107 was significantly reduced when PP2A/B55α incubated with the R1 peptide was used for dephosphorylation studies (*Figure 4B and C*). Repeating this experiment using a binding-deficient peptide (either R1 R621A or R2) showed dephosphorylation kinetics similar to that of the free PP2A/B55α, suggesting that the mutant peptide bound too weakly to compete for the substrate binding site on PP2A/B55α (*Figure 4B and C*). p107 R1 residues are critical for binding to PP2A/B55α and dephosphorylation.

To understand which R1 residues are critical for the p107:B55α interaction, we performed binding competition assays. In this assay, we use increasing concentrations of R1 peptide to compete for B55α binding (*Figure 5A*). This peptide was able to compete with a p107 R1R2 construct. As expected, using different peptides, such as R2 or small spacer peptides, did not show this effect. We performed this binding competition assay using endogenous PP2A/B55α from cellular extracts and observed no difference, highlighting that the source of PP2A/B55α did not matter (both replicates are quantitated in *Figure 5A*, right).

Next, we generated a large battery of R1 peptides – all with single or multiple Ala substitution – and performed this binding competition assay. Our data clearly showed that three triple-mutant peptides were unable to bind PP2A/B55α (*Figure 5B*). These data showed that the main R1 interaction residues must be within p107 M618-R626. Next, we tested the single Ala substitution peptides for p107 M618-R626 to identify all residues necessary for p107 binding to PP2A/B55α. These experiments showed that p107 residues H619, R621, V622, and V625 are critical for B55α binding as R1 peptides with H619A, R621A, V622A, and V625A bound poorly to PP2A/B55α (*Figure 5C*, *Figure 5—figure supplement 1*, top). Moreover, to confirm the requirement of each one of these residues for binding to B55α, we generated point mutants and tested them in binding assays. H619A, R621A, V622A, and V625A dramatically reduce binding to B55α holoenzymes (*Figure 5—figure supplement 1*, bottom). These experiments defined a putative SLiM binding motif for p107 – HxRVxxV – with PP2A/B55α.

To assess the relationship between binding and dephosphorylation of p107 by PP2A/B55α directly, we measured dephosphorylation of wildtype (wt) pS615-R1 or mutant (mt) variant (H619A or R621A) phosphopeptides (all phosphopeptides acted as their unphosphorylated counterparts in competitive binding assays, *Figure 5C* and *Figure 5—figure supplement 1*, top). *Figure 5D* shows that the mutant phosphopeptides were dephosphorylated by purified PP2A/B55α at a slower rate compared to the wildtype pS615-R1 phosphopeptide. These data provide us with direct evidence of delayed kinetics of PP2A/B55α-mediated dephosphorylation of a p107-derived substrate when binding is impeded.

Altogether, these binding and enzymatic assay data support a mechanism whereby the p107-based putative *HxRVxxV* SLiM docks at the mouth of the B55α groove (between blades 3 and 4 of the β-propeller) and the 'SP' site within the active site of the PP2A/C catalytic subunit.

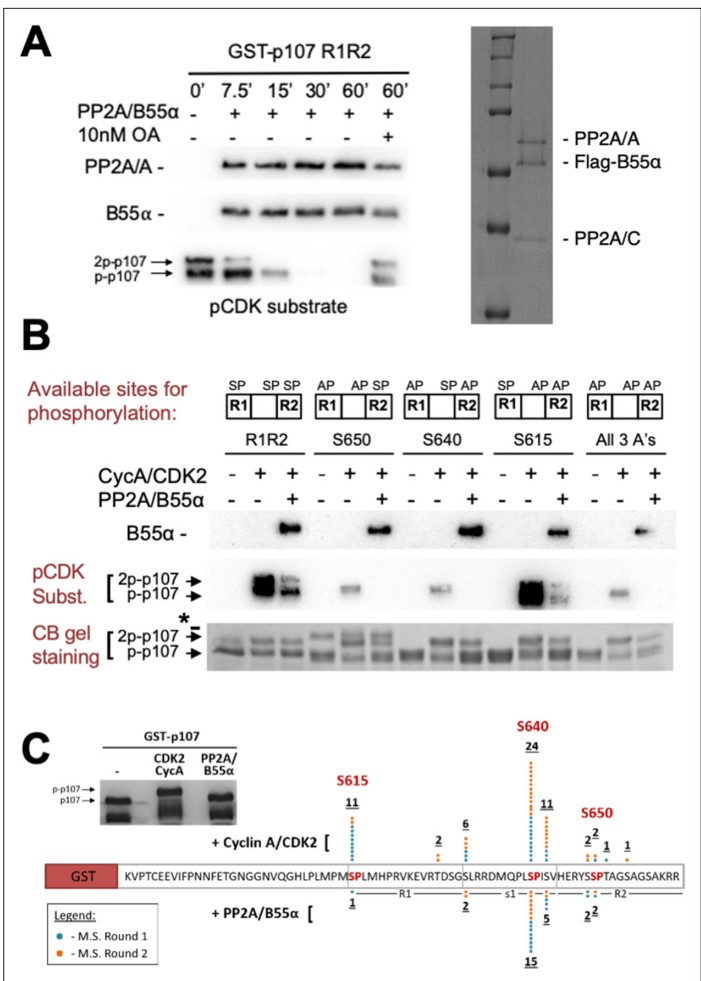

**Figure 3.** Combination of in vitro enzymatic assays and mass spectrometry identified S615 on R1 of p107 as the major site of PP2A/B55α-mediated dephosphorylation. (**A**) Dephosphorylation of GST-p107 R1R2 was performed using approximately 10 ng purified PP2A/B55α. The indicated time points were collected and samples were resolved via SDS-PAGE. Proteins were detected with anti-PP2A/A, anti-B55α, and anti-pCDK substrate [(K/H) pSP]. Representative Coomassie Blue-stained gel depicting affinity-purified PP2A/B55α holoenzymes is shown on right. (**B**) In vitro phosphorylation and dephosphorylation of GST-p107 R1R2 with single SP sites available were performed using 0.25 µg recombinant cyclin A/CDK2 and approximately 10 ng PP2A/B55α (each for 1 hr, respectively). Proteins were resolved via SDS-PAGE and detected by Coomassie Blue gel staining and western blotting using anti-B55α and pCDK substrate antibodies. Note basal levels of pCDK substrate signal in all '+ CycA/CDK2' lanes, which indicates phosphorylation on CDK2 itself. Relevant proteins and p107 phosphorylated species are indicated (a bacterial contaminant in the S650 MT is indicated with an asterisk). (**C**) Representative Coomassie Blue-stained PAGE used for mass spectrometry, with schematic summarizing the findings from two independent rounds of mass spectrometry analyses, is shown.

The online version of this article includes the following figure supplement(s) for figure 3:

**Source data 1.** Uncropped blots and Coomassie-stained gels for *Figure 3A-C*.

**Figure supplement 1.** Experiments performed to determine optimal GST-p107 R1R2 phosphorylation parameters using purified cyclin A/CDK2.

**Figure supplement 1—source data 1.** Uncropped Coomassie-stained gel, Phosphorimager exposure and western blots for *Figure 3—figure supplement 1*.

## B55α D197 plays a critical role in p107 recruitment

Next, we performed our binding and dephosphorylation experiments in cells. Using co-transfection followed by immunoprecipitation assays, we show that GFP-p107-R1, but not mutant GFP-p107-R1 (H/AxR/AV/AxxV/A variant), immunoprecipitates both endogenous B55α and exogenous Myc-B55α

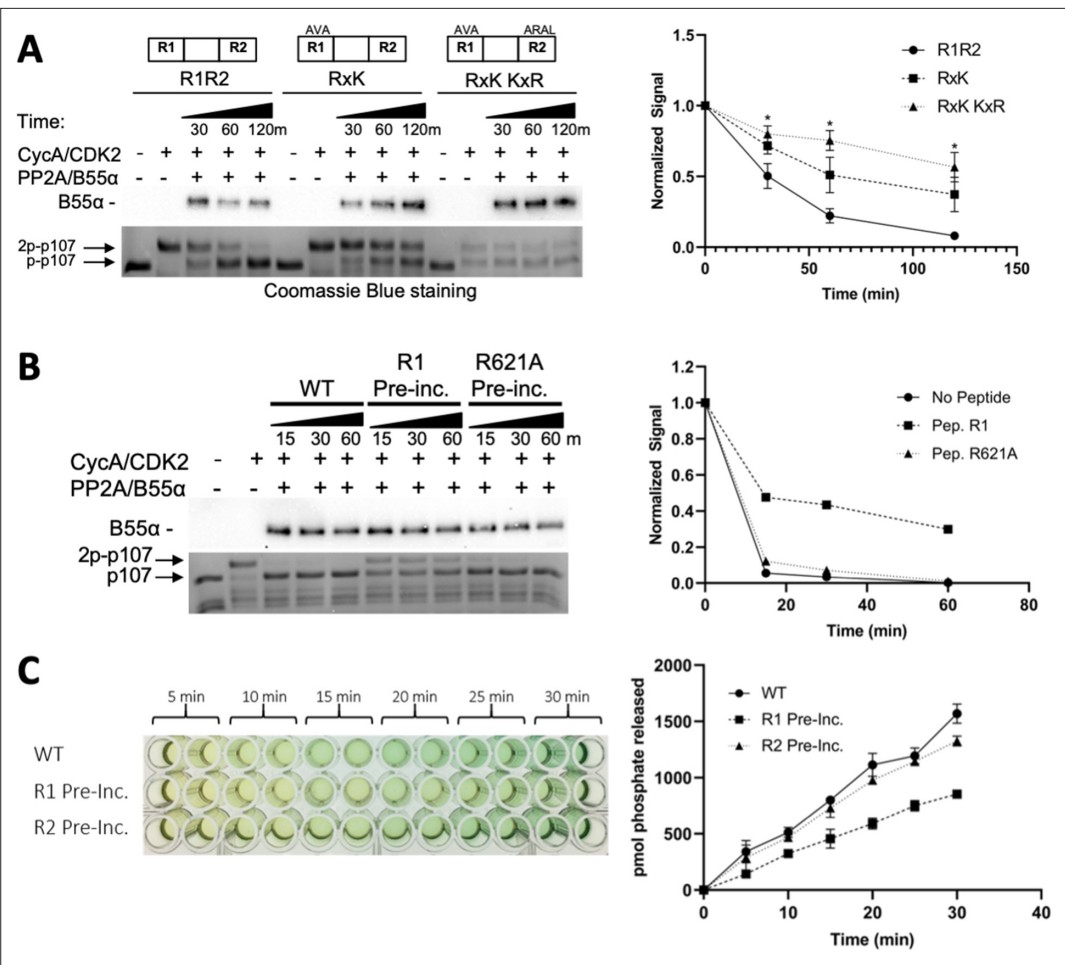

**Figure 4.** Residues critical for B55α/PP2A binding to p107 are critical for p107 spacer dephosphorylation. (**A**) Approximately 10 ng of purified PP2A/B55α was used to dephosphorylate wildtype GST-p107 R1R2 or mutant constructs (GST-p107 R621A K623A and GST-p107 RxK K657A R659A) in a time-course experiment. Proteins were resolved via SDS-PAGE and detected by Coomassie Blue staining or western blotting using anti-B55α antibodies. Quantifications of the 'phospho'-p107 band for each construct tested were performed using ImageJ and plotted as a function of time, with statistics shown above. (**B**) Representative assay in which purified PP2A/B55α was preincubated with either wildtype R1 peptide or R621A mutant R1 peptide and then used in time-course dephosphorylation assays using GST-p107 R1R2 (native enzyme was used as a positive control for dephosphorylation). Proteins were resolved via SDS-PAGE and detected by Coomassie Blue. Quantification is shown on right. (**C**) Purified PP2A/B55α was preincubated with either wildtype R1 or R2 peptide and then used in Malachite Green Phosphatase Assay with a p107-derived phosphopeptide (in which S615 is the available phosphosite). The indicated time points were collected and absorbance was read at 600 nm by microplate reader (quantification is shown on right).

The online version of this article includes the following figure supplement(s) for figure 4:

**Source data 1.** Uncropped Coomassie-stained gels and blots for *Figure 4A and B*.

---

together with PP2A/A (*Figure 6A*). As expected, GFP-R1 did not immunoprecipitate B55α-D197K (*Figure 6A*). This confirms that R1 is sufficient to form a complex with the PP2A/B55α holoenzyme in cells and that this is dependent on B55α D197. The specificity of the GFP-p107-R1 fusion protein pulldown was additionally validated by MS, which resulted in the identification of B55α and PP2A/A (*Figure 6B*). Therefore, the p107 R1 domain contains a highly specific SLiM for B55α family members. To determine if B55α modulates the phosphorylation state of CDK2 sites on p107 in cells, we transfected 293T cells with Flag-p107 alone or in combination with either wt B55α or mt B55α-D197K. *Figure 6C* shows that wt B55α expression, but not mt B55α-D197K, leads to reduced phosphorylation of p107 on sites recognized by the CDK substrate antibody, which include p107 S615 (this is the

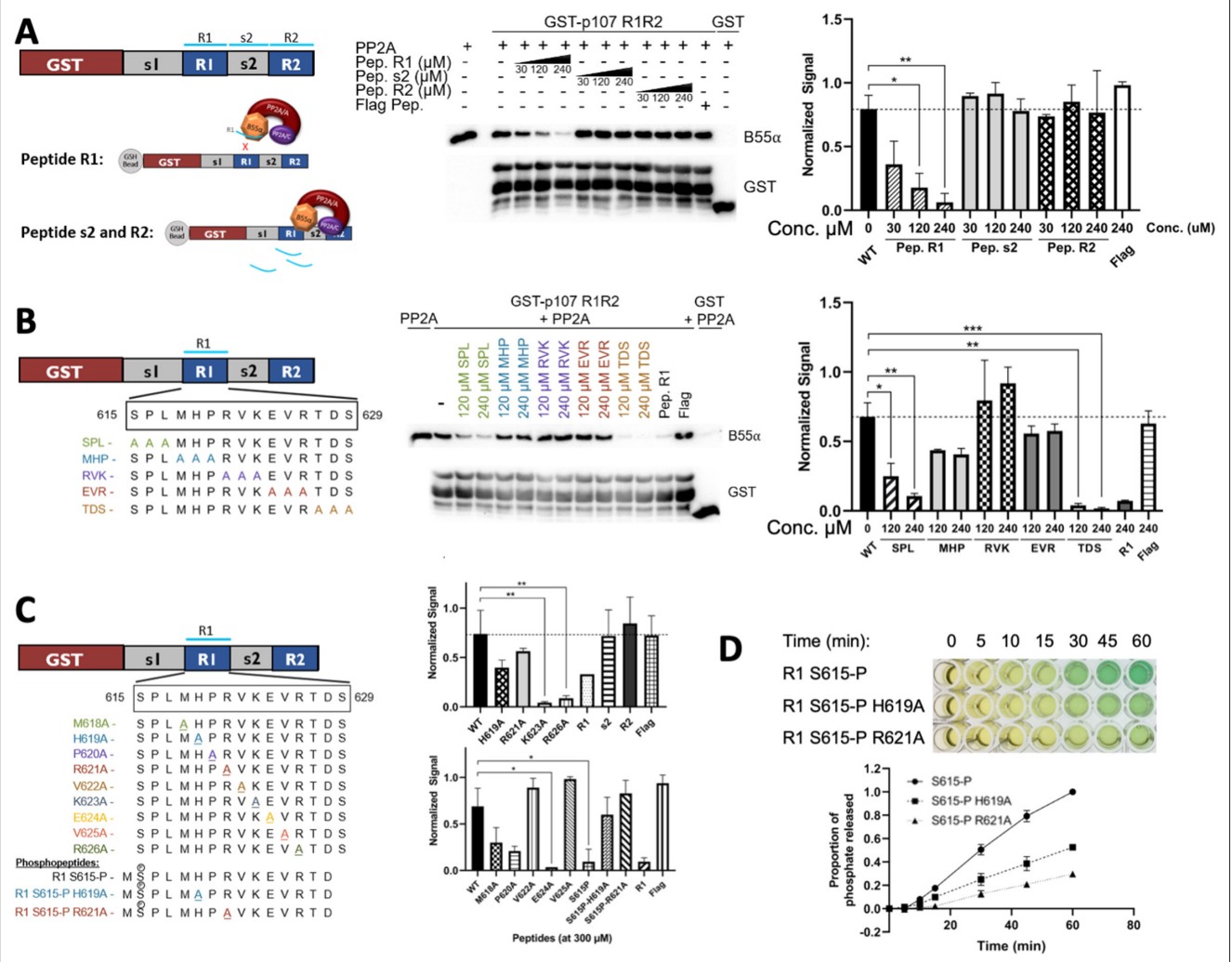

**Figure 5.** Identification of critical residues within the central 9-mer stretch of the 'R1' region of p107 for binding to B55α/PP2A (SPxxHxRVxxV). (**A**) PP2A/B55α purified from 293T-Flag-B55α cells was preincubated with synthetic p107 peptides and then used in pull-down assays with GST-p107 R1R2 constructs. Proteins were resolved via SDS-PAGE and detected via western blotting using anti-B55α and GST antibodies. An independent replicate experiment was performed using 293T cell lysates as source of PP2A/B55α with comparable results. Quantification of B55α pulled down relative to the wildtype pulldown was performed for both replicates using ImageJ, with statistics shown above. (**B, C**) Pulldowns were performed using purified PP2A/B55α and GST-p107 R1R2 as above with preincubations using mutant p107-derived synthetic peptides (both scanning triple-mutants and point mutants) as well as wildtype phosphopeptides. Proteins were resolved via SDS-PAGE and detected via western blotting using anti-B55α and GST antibodies. Quantifications were performed as above. (**D**) Purified PP2A/B55α was used in time-course Malachite Green Phosphatase Assay using wildtype or mutant p107-derived phosphopeptides as substrates of dephosphorylation. The indicated time points were collected and absorbance was read at 600 nm by microplate reader (quantification of duplicate assays is shown below).

The online version of this article includes the following figure supplement(s) for figure 5:

**Source data 1.** Uncropped upper membrane (anti-B55α) and lower membrane (anti-GST) western blots for *Figure 5A and B*.

**Source data 2.** Uncropped upper membrane (anti-B55α) and lower membrane (anti-GST) western blots for *Figure 5B, C* (three replicates) were used to quantitate peptide competition of B55α binding to GST-R1R2.

**Figure supplement 1.** Identifying esential rediues in the R1 region of the p107 spacer that affect binding to B55α, using peptide competition and GST puldown assays.

**Figure supplement 1—source data 1.** Uncropped western blot images for panel B.

only site recognized by this antibody in the p107 spacer, *Figure 3B*). To determine if the p107 spacer SLiM is required to modulate the phosphorylation state of S615 in full-length p107, we compared the effect of co-transfection of WT and D197K B55α on S615 phosphorylation of WT and SLiM mutant p107 ($_{619}$HxRVxxV$_{625}$ substituted by $_{619}$AxAAxxA$_{625}$). *Figure 6D* shows that the binding of WT B55α to

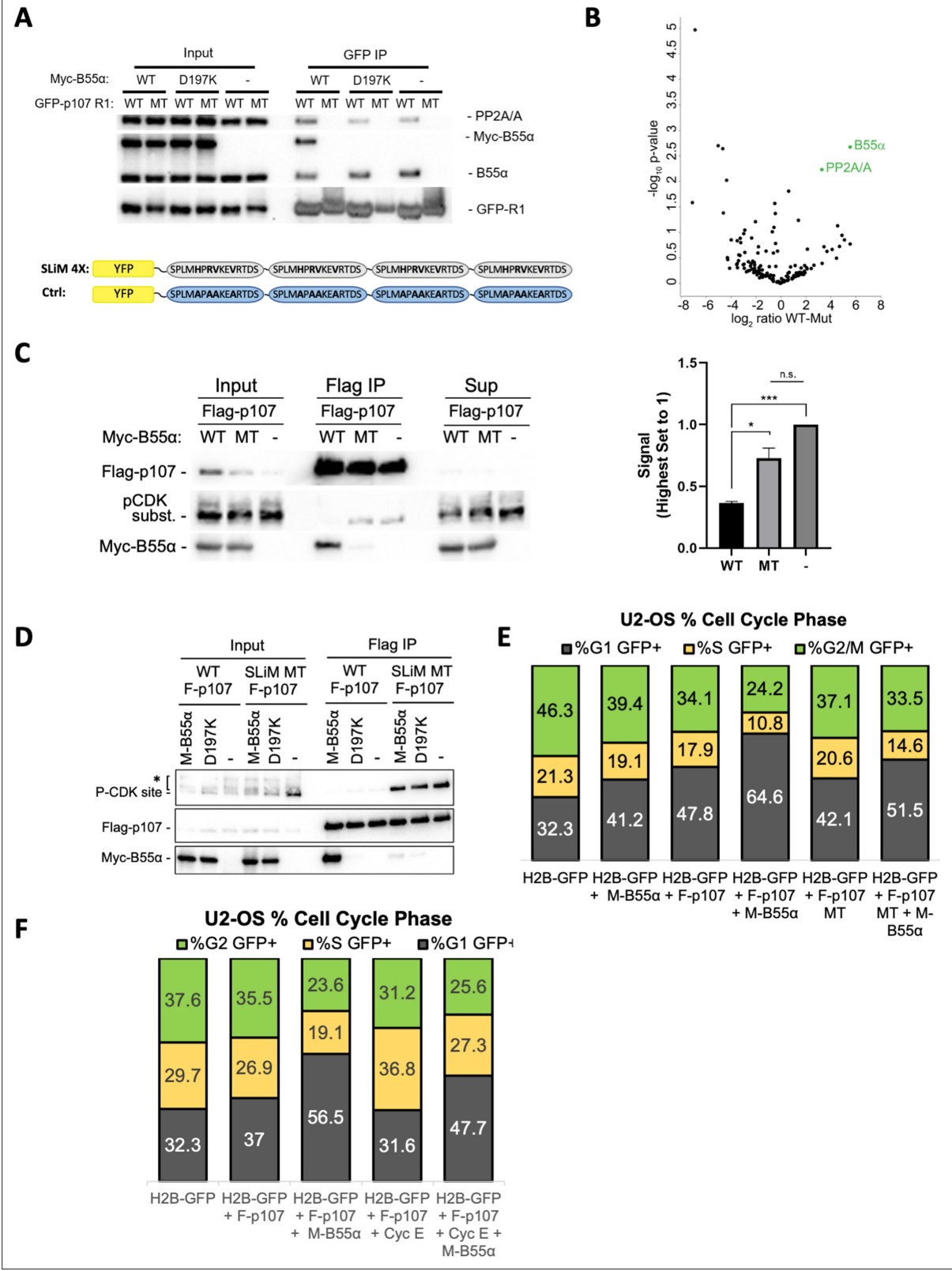

**Figure 6.** p107 R1 interaction with the B55α holoenzyme in cells depends on sites required for binding and dephosphorylation in vitro. (**A**) GFP-p107 R1 wildtype and mutant constructs were co-transfected with Myc-B55α wildtype and B55α-D197K mutant constructs into 293T cells and used for IPs with anti-GFP agarose conjugate. Input lysates and IPs were resolved via SDS-PAGE, and proteins were detected using anti-PP2A/A, anti-B55α, and anti-GFP. Schematic of WT and MT GFP-p107 R1 constructs is shown below. (**B**) Volcano plot depicts the differences of protein abundances in YFP-SLIM4X and

*Figure 6 continued on next page*

*Figure 6 continued*

YFP-Ctrl pulldowns (*Figure 6—source data 1*). (**C**) Flag-p107 was co-transfected with Myc-B55α wildtype and B55α-D197K mutant constructs into 293T cells and used for IPs with anti-Flag agarose conjugate. IPs and input/supernatant lysates were resolved via SDS-PAGE, and proteins were detected using anti-Flag and anti-pCDK substrate antibodies. (Right) Quantification of pCDK substrate signal (relative to Flag) is shown, with statistics shown above. (**D**) (Left) WT and short linear motif (SLiM) MT Flag-p107 were co-transfected with Myc-B55α wildtype and B55α-D197K mutant constructs into 293T cells and analyzed as in (**C**). (**E**) U-2 OS cells were co-transfected with the indicated plasmids, and the percent of cells in G1, S, and G2/M were determined by measuring DNA content via FACS/PI analyses of GFP-positive cells. (**F**) U-2 OS cells were co-transfected with the indicated plasmids and analyzed as in (**D**, right).

The online version of this article includes the following figure supplement(s) for figure 6:

**Source data 1.** Uncropped upper membranes for *Figure 6A* and replicate experiments of endogenous B55α interaction with wildtype, but not mutant, GFP-R1-SLiM.

**Source data 2.** Mass spectrometry dataset used to generate the volcano plot shown in *Figure 6*.

**Source data 3.** Uncropped upper and lower membranes for *Figure 6C and D* (representative replicate 1 was selected for the C and D panels).

**Figure supplement 1.** U-2 OS cells were co-transfected with the indicated plasmids, lysates were immunoprecipitated with anti-Flag agarose conjugate, and the indicated co-immunoprecipitated proteins were detected by western blot analyses.

the p107 SLiM mutant is drastically diminished and this is accompanied by a dramatic increase in the phosphorylation of p107 S615. As expected, B55α$_{D191K}$ does not bind WT or MT p107, nor affects the phosphorylation state. Therefore, the p107 SLiM is essential to maintain steady-state phosphorylation of at least p107 S615, and thus we can infer that B55α must be critical for the temporal regulation of at least this site. We next determined if disruption of the SLiM affects p107 G1 cell cycle suppression activity, and if this is dependent on B55α. U-2 OS cells were co-transfected with H2B-GFP or H2B-GFP with Flag-p107 or the Flag-p107 SLiM MT alone or in combination with Myc-B55α. DNA content was analyzed by PI/FACS analysis of GFP-positive cells. Consistent with previous reports (*Farkas et al., 2002*), Flag-p107 expression led to a noticeable increase in the fraction of cells in G1 (*Figure 6E*). Importantly, this effect was drastically increased by coexpression of Myc-B55α, which had a smaller effect on its own. The effect of the Flag-p107 SLiM mutant in G1 was clearly diminished relatively to WT Flag-p107 even with co-transfected Myc-B55α. Altogether, these data strongly suggest that B55α-mediated dephosphorylation of S615 in cells is dependent on specific residues C-terminal from the target dephosphorylation site that mediate contacts with B55α D197 and/or neighboring residues in B55α, and that the SLiM is critically important for modulation of the cell cycle suppressing function of p107.

To elucidate if the effect of B55α on p107 G1 suppressor functions is dependent on E2F4, the preferred E2F partner of p107, U-2 OS cells, was co-transfected with Flag-p107 alone or in combination with HA-E2F4, Myc-B55α, and/or cyclin E. We selected cyclin E because cyclin E/CDK2 is thought to cooperate with cyclin D/CDK4 to disrupt E2F complexes in various cell types (*Calbó et al., 2002*; *Lundberg and Weinberg, 1998*), and our in vitro data show that it can be phosphorylated by CDK2 (*Figure 3*), whereas p107 S615 is not a target of CDK4 (*Farkas et al., 2002*). Cell cycle analyses showed that cyclin E clearly reduces the p107-induced G1 arrest, and this effect is counteracted by Myc-B55α (*Figure 6F*). However, co-transfection of Myc-B55α and/or cyclin E had little effect on the abundance of Flag-p107/HA-E2F4 complexes (*Figure 6—figure supplement 1*). Altogether, it suggests that B55α promotes p107 activation and cell cycle arrest at least in part by opposing cyclin E/CDK2 function likely by mechanisms independent of E2F4. Of note, we also observed shifts in E2F4 protein isoform migration with Myc-B55α co-transfection, which may indicate that E2F4 is also dephosphorylated, perhaps as a result of complex formation with p107.

## A model for p107 recruitment by PP2A/B55α

Based on all the established data, we created a model to describe the recruitment of p107 by PP2A/B55. This model is based on the following premises: (1) using HxRVxxV as a ruler, we assumed that either H619$_{p107}$ or R621$_{p107}$ interacts with D197$_{B55α}$; and (2) as we confirmed that pS615$_{p107}$ is dephosphorylated by PP2A/B55α, pS615$_{p107}$ must be placed into the PP2A active site. In order to place pS615$_{p107,}$ it became more likely that R621$_{p107}$ interacts with D197$_{B55α}$ for simple distance reasons. This meant that pSPMLHPR will need to connect the PP2A active site and D197$_{B55α}$. To identify possible existing confirmations of such a peptide, we searched the PDB database for peptide fragments from existing proteins structure using the sequence EPxxxPR (pS was exchanged to the E mimic to

identify more peptides). Next, we examined the distribution of distances between the oxygen atoms of the $OE_1/OE_2$ atoms of the E side chain and the $NH_1/NH_2$ atoms of the R side chain, and compared these to the distance between E in the PP2A active site and $R621_{p107}$ bound to $D197_{B55\alpha}$, which is 27.5 Å (*Figure 7A and B*, *Figure 7—figure supplement 1*, top). The 520 identified peptides with the sequences xEPxxxPRx have distances between 15 and 25 Å (*Figure 7—figure supplement 1*), indicating that p107 pSPMLHPR must form a highly extended structure when bound to PP2A/B55. Lastly, we used an extended EPxxxPR peptide, placed it into the PP2A/B55α complex structure, mutated the EPxxxPR sequence to the p107 sequence (MpSPMLHPRV), and refined these structures using FlexPepDock (*London et al., 2011*; *Raveh et al., 2010*; *Raveh et al., 2011*).

In the best PP2A/B55α/p107 model, the $NH_2$ side chain of $R621_{p107}$ is 3.2 Å apart from $D197_{B55\alpha}$, while the $OE_1/OE_2$ of E615 (pS615 mimic) is 4.4 and 5.1 Å apart from the two $Mn^{2+}$ ions at the PP2A active site, respectively (*Figure 7C–E*). These distances are slightly longer than expected for this type of interactions, but replacing E615 with pS615 shows consistency with active site binding of PPPs (*Figure 7F*; *Goldberg et al., 1995*). Taken together, our model shows that p107 pSPLMHPRV can bind in a highly extended fashion to the PP2A/B55α holoenzyme, in full agreement with our experimental data.

## A derived p107 -pSPxxHxRVxxV- SLiM is conserved in other substrates and functional validated in TAU

Peptides known to abrogate TAU and MAP2 binding to B55α have been reported (*Sontag et al., 2012*). We noticed that TAU and MAP2 and the conserved p107 family member, p130, contain residues that align with our defined p107 SLiM, generating a consensus sequence, p[**ST**]-**P**-x(4,10)-[**RK**]-**V**-x-x-[**VI**]-**R** (*Figure 8A*) and other cellular IDPs, where the phosphosite is 5–10 residues amino terminal from the conserved residues in the binding motif. To functionally validate the conservation of the p107 SLiM, we selected TAU, a well-known B55α/PP2A substrate, which has been shown to require acidic residues on the B55α surface for binding, but any other residues that form a defined SLiM are unknown. Therefore, we generated a phospho-TAU peptide encompassing the putative conserved SLiM and a variant peptide with the conserved residues mutated to Ala. *Figure 8B* shows that disruption of critical residues within the newly defined B55α substrate SLiM completely blocks TAU dephosphorylation.

More degenerate versions of the SLiM (p[ST]-P-x(4,10)-[RK]-[VIL]-x-x-[VILM]-[RK]) considering conserved amino acid substitutions identify 309 hits in 302 sequences, 1.5% of the proteins in the Swiss Human Proteome. However, as the last residue in the SLiM contributes minimally to p107:B55α binding despite its alignment with TAU (*Figures 5C and 8A*), we determined the frequency of the potential SLiM sequence (p[ST]-P-x(4,10)-[RK]-[VIL]-x-x-[VILM]-x), which retrieved 2831 hits in 2422 sequences (13.9% of the proteome) (*Figure 8—figure supplement 1*). Of note, considering that the SLiM would have to be conserved, in an IDR and controlling dephosphorylation of a phosphosite(s), the frequency of a genuine SLiM is predictably much smaller. Thus, we rationalized that if this B55α substrate SLiM is functional in a significant fraction of B55α substrates, SLiM occurrence will be expected to relatively increase in datasets of B55α interactors or datasets where modulation of B55α expression results in phosphoproteome changes. This is in fact what we have observed. *Figure 8—figure supplement 1* shows that the fraction of proteins with the SLiM greatly increases among a dataset of B55α interactors (*Hertz et al., 2016*) and potential in vitro B55α mitotic substrates (*Kruse et al., 2020*). This enrichment indicates that many substrates and B55 interactors potentially use this SLiM-based mechanism.

## Discussion

PP2A is a key Ser/Thr protein phosphatase that requires its B regulatory subunits to specifically recruit substrates. PP2A B-subunits are proteins that adopt specific folds and thus allow for specific substrate recruitment. Much work over the last few years has highlighted how the B56 regulatory subunit recruits its substrates. A central, conserved groove region in B56 provides the necessary and sufficient interaction site for the substrate recognition sequence (SLiM site) – LxxIxE (*Hertz et al., 2016*; *Wang et al., 2016*). Much less is known for B55, despite the fact that it is the most abundant B subunit and targets a myriad of key cellular substrates. Previous reports highlighted the possibility that B55α functions

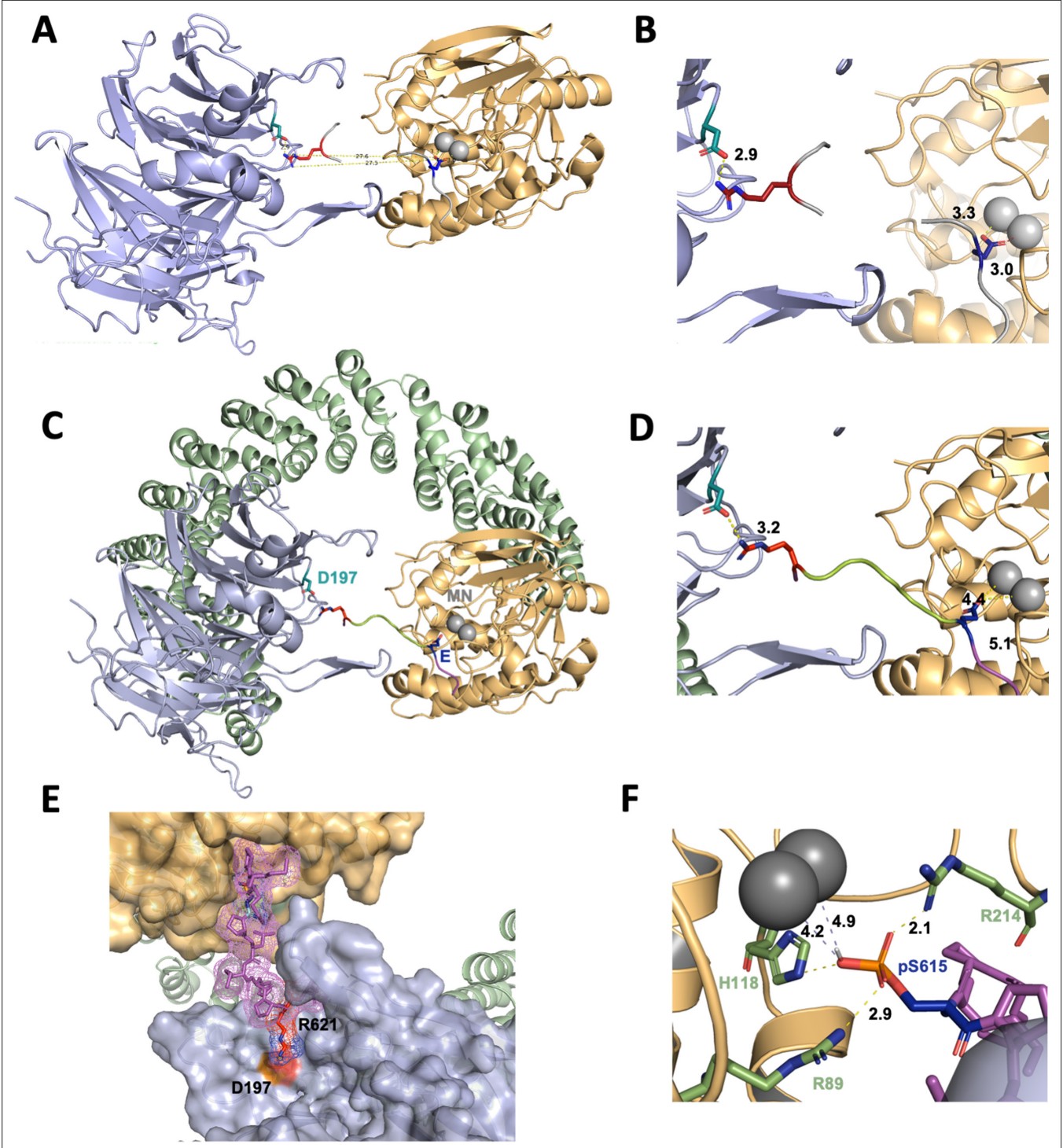

**Figure 7.** A computational model of the p107 phosphopeptide (613–622) binding B55α and the active site of PP2A/C is consistent with p107 spacer contacts to B55α as determined by NMR spectroscopy. (**A**) The 'two-fragment' model depicting B55α and PP2A/C with two modeled peptide fragments. Distances between the OE$_1$/OE$_2$ atoms of Glu and the NH$_1$/NH$_2$ atoms of Arg are shown in center. (**B**) A closer view of 'two-fragment' model highlighting the distances between Arg and D197 of B55α and between the Glu residue and the Mn$^{2+}$ ions within PP2A/C active site. (**C, D**) Peptide model depicting the best alignments with the two reference fragments and the best distances to the Mn$^{2+}$ ions and D197. A closer view highlighting the distances between the NH$_2$ of Arg residue and D197 of B55α, and the Glu residue with the Mn$^{2+}$ ions, is shown in (**C**). (**E**) Close-up of model in (**C, D**) showing the peptide side chains and contacts to B55α surface. (**F**) Close-up of pSer-621 (substituted for the Glu residue) and residues critical for phosphate coordination (R630, H559, R655). Distances between residues where H-bonding is predicted to occur are shown.

*Figure 7 continued on next page*

*Figure 7 continued*

The online version of this article includes the following figure supplement(s) for figure 7:

**Source data 1.** PyMOL session source data for *Figure 7A*.

**Source data 2.** PyMOL session source data for *Figure 7B*.

**Source data 3.** PyMOL session source data for *Figure 7C and D*.

**Source data 4.** PyMOL session source data for *Figure 7E*.

**Source data 5.** PyMOL session source data for *Figure 7F*.

**Figure supplement 1.** Step details to generate a computational model of the p107 phosphopeptide (613-622) binding B55α and the active site of PP2A/C.

**Figure supplement 1—source data 1.** Distances between $OE_{1/2}$ and $NH_{1/2}$ for 520 peptide structures with the consensus sequence EPXXXPR retrieved from PDB.

**Figure supplement 1—source data 2.** R script file to generate density plots for *Figure 7—figure supplement 1A and B* .

**Figure supplement 1—source data 3.** PyMOL session source data for *Figure 7—figure supplement 1B*.

**Figure supplement 1—source data 4.** Distances of $OE_{1/2}$ and $NH_{1/2}$ between each peptide and two reference fragments for *Figure 7—figure supplement 1C* .

**Figure supplement 1—source data 5.** PyMOL session source data for *Figure 7—figure supplement 1D*.

**Figure supplement 1—source data 6.** PyMOL session source data for *Figure 7—figure supplement 1E*.

**Figure supplement 1—source code 1.** Model code.

**Figure supplement 2.** PP2A/A scaffold flexibility upon binding to the catalytic subunit and B subunits of the four distinct holoenzymes.

**Figure supplement 2—source data 1.** PyMOL session source data for *Figure 7—figure supplement 2A*.

**Figure supplement 2—source data 2.** PyMOL session source data for *Figure 7—figure supplement 2B*.

**Figure supplement 2—source data 3.** Scaffold variation source data without regulatory subunit for *Figure 7—figure supplement 2C*.

**Figure supplement 2—source data 4.** Scaffold variation source data with B56 subunits for *Figure 7—figure supplement 2D*.

---

differently with the potential presence of distinct substrate recruitment sites. Specifically, sites of the acidic surface of B55α that were required for Tau dephosphorylation (*Xu et al., 2008*) appeared dispensable for p107 binding (*Jayadeva et al., 2010*). Here, we used a broad range of approaches – from molecular to biochemistry to cellular – to understand how PP2A/B55α recruits specific substrates using p107 as a model substrate.

We show that p107 S615 (CDK kinase phosphorylation site) is a specific B55α/PP2A holoenzyme dephosphorylation site. p107 residues H619, R621, V622, and V625 within the p**S**Pxxx**H**x**RV**xx**V** motif are critical for binding to B55α in vitro and in cells. Next, we showed that these residues bind to a B55α surface groove that is defined by residues D197 and L225. Importantly, our results show that this B55α surface is also important for recruitment of pRB and KSR, further suggesting that these sites are likely the key substrate recruitment sites for a variety of substrates. However, pRB and KSR differ in the requirement of other conserved residues present within the B55α groove, strongly suggesting that while substrate specificity is determined by the groove, it may use different residues to accommodate a variety of substrates. With this regard, we have found that peptides known to abrogate Tau and MAP2 binding to B55α (*Sontag et al., 2012*) contain residues that align with our proposed SLiM consensus sequence, **p[ST]-P-x(4,10)-[RK]-V-x-x-[VI]-R,** that is also found in p130/RBL2 (*Figure 8A*) and other cellular IDPs, where the phosphosite is 5–11 residues amino terminal from the conserved residues in the binding motif. Strikingly, purified B55α/PP2A holoenzymes dephosphorylate the wild-type TAU peptide, but fail to dephosphorylate a mutant variant that lacks the conserved residues. It appears that the number of residues between the conserved residues that contact the groove and the phosphorylation site varies but can be very small (5 residues in p107 vs. 7 residues in TAU and 11 in MAP2), which suggests a role for the SLiM in phosphosite presentation. In contrast, recent work on the B56 SLiM suggests that B56 SLiM-containing proteins are direct substrates as well as scaffolds to facilitate the recruitment of proteins to B56/PP2A for dephosphorylation (*Kruse et al., 2020*). While there is some correlation between the distance of the B56 SLiM and the site of dephosphorylation and the rate of dephosphorylation, many B56-dependent phosphorylation sites are located on non-SLIM-containing proteins indicating B regulatory subunit-specific dephosphorylation mechanisms. Computational search for a degenerate version of the SLiM ([ST]-P-x(4,10)-[RK]-[VIL]-x(2)-[VILM])

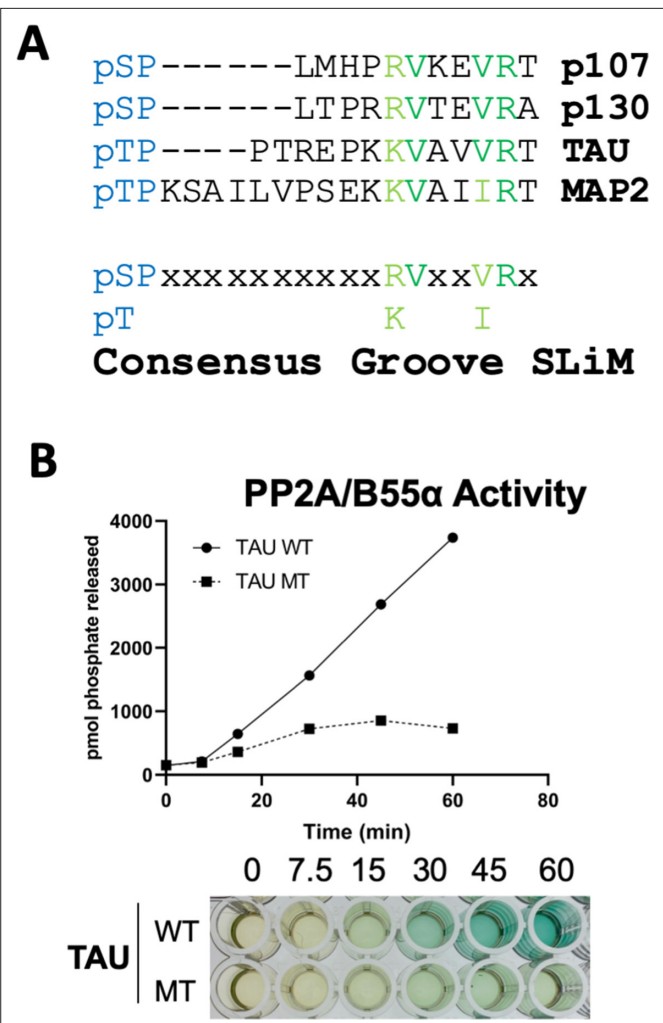

**Figure 8.** A derived p107 -pSPxxHxRVxxV- short linear motif (SLiM) is conserved in other substrates and functional validated in TAU. (**A**) Schematic of our proposed hypothetical consensus groove SLiM, p[ST]-P-x(4,10)-[RK]-V-x-x-[VI]-R, for TAU, MAP2, and the conserved p107 family member, p130, each of which contain residues that align with our defined p107 SLiM. (**B**) Time-course Malachite Green Phosphatase Assay using a wildtype phospho-TAU peptide encompassing the putative conserved SLiM and a variant peptide with the conserved residues mutated to Ala. (Above) Quantification of phosphatase assay is shown below.

The online version of this article includes the following source data and figure supplement(s) for figure 8:

**Figure supplement 1.** Degenerate short linear motif (SLiM) search in the proteome vs. datasets enriched for B55 interactor and potential substrates.

**Figure supplement 1—source data 1.** Table of proteins containing the [ST]-P-x(4,10)-[RK]-[VIL]-x(2)-[VILM] sequence in a dataset of B55α interactors (**Hertz et al., 2016**) and potential in vitro B55α mitotic substrates (**Kruse et al., 2020**).

demonstrated a sharp enrichment for the presence of SLiM sequences among B55α interactors and mitotic in vitro substrates, indicating that the presence of the SLiM is not random. If these putative B55α substrate SLiMs are conserved through evolution in these proteins and found in IDRs, they may contribute to B55α binding and/or substrate recognition.

Moreover, we also identified a second region of p107, which we termed R2, that contributes to B55α binding. This R2 region includes a higher density of positively charged residues that enhance binding most likely via charged:charged interactions ($R_{647}$, and $K_{657}$ and/or $R_{659}$). Indeed, this has also been recently shown for protein phosphatase 2B (PP2B; PP3; calcineurin) and the B56 regulatory subunit of PP2A, where these dynamic charged:charged interactions were shown to play a role in

substrate specificity (*Hendus-Altenburger et al., 2019*; *Wang et al., 2020*). Thus, such interactions, which have been recently recognized to be critical for the interaction of IDPs with their target proteins and allow for increased entropy, are likely important for substrate recruitment and specificity by different phosphatases.

Lastly, we leveraged our mutagenesis data to understand the p107 R1 binding site on B55α by generating a model how p107 R1 engages simultaneously with B55α and the PP2Ac active site. Our computational model based on structural data of peptides with the conserved key residues on p107 needed for B55α binding shows the feasibility of simultaneous binding of both the active site of PP2A/C and D197 on the B55α groove. However, the model peptide is significantly extended, a conformation that would not be favored for catalysis without additional bending of the scaffold subunit. Analysis of the flexibility of the PP2A/A scaffold upon binding to the catalytic subunit, and B subunits of the four distinct holoenzymes, confirms that subunit binding has a major effect on PP2A/A bending (*Figure 7—figure supplement 2*). The dependency of the conformational flexibility of the scaffold subunit on the nature of the B subunit bound strongly suggests that substrate binding should result in additional relative movement of PP2A/A HEAT-repeats to promote catalysis.

Taken together, our detailed molecular and cellular study highlights how PP2A/B55α recruits substrates and how these substrates engage with the PP2A active site, and uncovers the top groove of B55α as a central hub for substrate discrimination. In addition, this work will also facilitate identification and validation of new substrates based on the presence of variant SLiMs, a key step towards understanding B55α/PP2A's role in multiple cellular processes.

# Materials and methods

## Key resources table

| Reagent type (species) or resource | Designation | Source or reference | Identifiers | Additional information |
|---|---|---|---|---|
| Antibody | CDK2 (rabbit polyclonal) | Santa Cruz | sc-163 | WB (1:1000) |
| Antibody | Cyclin A (mouse monoclonal) | Santa Cruz | sc-271682 | WB (1:1000) |
| Antibody | Flag (mouse monoclonal) | Sigma | A8592-.2MG | WB (1:2500, 1:10,000) |
| Antibody | Flag (mouse monoclonal) | GenScript | A00187 | IP (1 µg/mL) |
| Antibody | GFP (rabbit monoclonal) | CST | 2956S | WB (1:1000) |
| Antibody | GST (mouse monoclonal) | Santa Cruz | sc-138 | WB (1:1000) |
| Antibody | PP2A Aβ subunit (goat polyclonal) | Santa Cruz | sc-6113 | WB (1:2000) |
| Antibody | PP2A B subunit (100C1) (rabbit monoclonal) | CST | 2290S | WB (1:2000) |
| Antibody | PP2A C subunit (1D6) (mouse monoclonal) | Millipore | 05-421 | WB (1:5000) |
| Antibody | Phospho-CDK Substrate Motif (rabbit monoclonal) | CST | 9477S | WB (1:1000) |
| Antibody | ECL Rabbit IgG, HRP-linked whole Ab (from donkey) | GE Healthcare | NA934V | WB/secondary antibody (1:10,000) |
| Antibody | HA (12CA5) (mouse monoclonal) | Roche/Sigma | 11583816001 | WB (1:500) |
| Antibody | ECL Mouse IgG, HRP-linked whole Ab (from donkey) | GE Healthcare | NA931V | WB/secondary antibody (1:10,000) |

*Continued on next page*

*Continued*

| Reagent type (species) or resource | Designation | Source or reference | Identifiers | Additional information |
|---|---|---|---|---|
| Antibody | Mouse anti-goat IgG-HRP | Santa Cruz | sc-2354 | WB/secondary antibody (1:10,000) |
| Antibody | Monoclonal ANTI-FLAG M2 antibody produced in mouse, ANTI-FLAG M2 Affinity Agarose Gel | Sigma | A2220 | IP (10 µL) |
| Antibody | Anti-c-Myc Agarose Affinity Gel antibody produced in rabbit (polyclonal) | Sigma | A7470 | IP (10 µL) |
| Antibody | GFP-Trap agarose beads | Chromotek | gta-10 | IP (5 µL) |
| Peptide, recombinant protein | Synthetic peptides for competition assays | Biomatik | Custom | Sequence variant described in this paper |
| Peptide, recombinant protein | DYKDDDDK peptide | GenScript | RP10586 | |
| Peptide, recombinant protein | Recombinant cyclin A/CDK2 | Thermo Fisher | PV3267 | |
| Peptide, recombinant protein | Purified recombinant B55α/PP2A holoenzyme | *Zhao et al., 2019* | | Trimeric Flag-B55α/PP2A holoenzyme purified from 293T cells. |
| Strain, strain background (*Escherichia coli*) | BL21-Gold (DE3) cells | Agilent | 230132 | To generate GST-Fusion proteins |
| Cell line (*Homo sapiens*) | 293T cells | ATCC | CRL-3216 | Transient transfections and source of cell lysates |
| Cell line (*H. sapiens*) | U-2 OS cells | ATCC | HTB-96 | Transient transfections |
| Cell line (*H. sapiens*) | Expi293F cells | Thermo Fisher | A14528 | Purification of recombinant B55α |
| Transfected construct (human) | Flag-B55α 293T cells | *Zhao et al., 2019* | | 293T cells stably transfected with pCPP-Flag-B55α and selected with puromycin |
| Commercial assay or kit | (AminoLink Plus Immobilization Kit) | Thermo Fisher Scientific | 44894 | |
| Commercial assay or kit | Ser/Thr phosphatase assay kit | EMD Millipore | 17-127 | |
| Commercial assay or kit | QuikChange II | Agilent | 200521 | |
| Recombinant DNA reagent | pBOS GFP-H2B plasmid | *Kanda et al., 1998* | | |
| Recombinant DNA reagent | pcDNA3.4-K-GFP-RP1B | This paper | | His6-green fluorescent protein-tag, a TEV cleavage |
| Recombinant DNA reagent | pTHMT | Peti and Page, *Protein Expr. Purif.* **51**, 1–10 (2007) | | N-terminal His6-Maltose Binding Protein (MBP)-tag, a TEV cleavage |
| Recombinant DNA reagent | pCPP-Flag-B55α | *Zhao et al., 2019* | | |

*Continued on next page*

*Continued*

| Reagent type (species) or resource | Designation | Source or reference | Identifiers | Additional information |
|---|---|---|---|---|
| Recombinant DNA reagent | pMSCV-puro-Myc-B55α | *Jayadeva et al., 2010* | | |
| Recombinant DNA reagent | pMSCV-puro-Myc-B55α variants | This paper | | Generated by site-directed mutagenesis (primer sequences provided in Appendix table) |
| Recombinant DNA reagent | pCDNA5/FRT/TO-GFP-p107-R1 | This paper | | Inserting wild-type or mutant p107-R1 gBlock Gene Fragments (IDT) into the pCDNA5/FRT/TO-GFP vector via the BamHI/Not sites |
| Recombinant DNA reagent | pGEX-2T-p107 spacer | *Jayadeva et al., 2010* | | |
| Recombinant DNA reagent | pGEX-2T-p107 spacer variants | This paper | | Generated by site-directed mutagenesis (primer sequences provided in Appendix table) |
| Recombinant DNA reagent | pCDNA5/FRT/TO-GFP-p107-R1 | This paper | | Inserting a gblock in pCDNA5/FRT/TO-GFP containing 4 copies of R1 |
| Recombinant DNA reagent | pCDNA5/FRT/TO-GFP-p107-R1-H/AxR/AV/AxxV/A | This paper | | Inserting a gblock in pCDNA5/FRT/TO-GFP containing four mutant copies of R1. (gblock sequences provided in Appendix table) |
| Recombinant DNA reagent | pSG5-puro-Flag-p107 | *Kurimchak et al., 2013* | | |
| Recombinant DNA reagent | pCMV-Flag-p107 | *Voorhoeve et al., 1999* | | |
| Recombinant DNA reagent | pCMV-HA-E2F4 | *Ginsberg et al., 1994* | | |
| Recombinant DNA reagent | pRC-cyclin E | Addgene | #8963 | |
| Software, algorithm | ConSurf | ConSurf *Ashkenazy et al., 2010* | | |
| Software, algorithm | ImageJ software | | | |
| Software, algorithm | FlowJo v10 software | BD Biosciences | v10.8 | |
| Software, algorithm | Clustal Omega | *Sievers et al., 2011* | | |

## Cell culture and cell lines

All cell lines were obtained from ATCC and cultured in DMEM supplemented with 10% FBS and 0.1% Penicillin-Streptomycin as described previously (*Jayadeva et al., 2010*) and tested for mycoplasma annually. For stable expression of Flag-B55α, 293T cells were transfected with pCPP-puro-Flag-B55α followed by puromycin selection for clone generation. Transient expression of Myc- and Flag-containing constructs was achieved using calcium phosphate transfection. Briefly, 5 µg plasmid DNA was added dropwise to 2× HEPES-buffered saline (HBS) solution (280 mM NaCl, 50 mM HEPES, 1.5 mM $Na_2HPO_4$, pH 7.05) with bubbling, followed by addition to cells treated with 25 mM chloroquine after a 30 min incubation period.

## Plasmids

Plasmids used or generated in this study are described in the Key resources table. pGEX-2T-GST-p107 spacer, pMSCV-puro-Myc-B55α, and pSG5-puro-Flag-p107 variant mutants were generated by site-directed mutagenesis using the QuikChange II Site-directed mutagenesis kit (Agilent) and the primers listed in the Appendix table and subsequently validated by Sanger sequencing.

## GST pull-down assays

GST-tagged constructs of interest were expressed in *Escherichia coli* bacteria and purified for use in pull-down assays. Briefly, 100 mL cultures of *E. coli* were treated with 0.25 mM isopropyl

β-D-thiogalactoside (IPTG) for 2 hr to induce expression of GST-fusion proteins. Cells were then harvested by centrifugation and resuspended in NETN lysis buffer (20 mM Tris pH 8, 100 mM NaCl, 1 mM EDTA, 0.5% NP-40, 1 mM PMSF, 10 μg/mL leupeptin) prior to sonication at 30% amplitude for 10 cycles. Supernatants were collected and incubated with glutathione beads for purification, followed by NETN buffer washes for sample clean-up. For pull-down assays, purified GST-p107 spacer and mutant constructs were incubated with 293T lysates for 3 hr or overnight at 4°C, followed by washes (5×) with complete DIP lysis buffer (50 mM HEPES pH 7.2, 150 mM NaCl, 1 mM EDTA, 2.5 mM EGTA, 10% glycerol, 0.1% Tween-20, 1 μg/mL aprotinin, 1 μg/mL leupeptin, 1 μg/mL Pepstatin A, 1 mM DTT, 0.5 mM PMSF) and elution with 2× LSB. Samples were resolved by SDS-PAGE and probed using antibodies against proteins of interest.

### Peptide competition assays

Synthetic peptides derived from the amino acid sequence of p107 were generated and used in competition assays with GST-tagged p107 R1R2 constructs (Biomatik). Briefly, purified PP2A/B55α holoenzymes were preincubated with 30–300 μM synthetic p107 peptides for 30 min at 37°C to facilitate interaction. These preincubated PP2A/B55α complexes were then incubated with GST-tagged p107 R1R2 constructs for 3 hr or overnight at 4°C, followed by washes (5×) with NETN lysis buffer and elution with 2× LSB. Proteins were resolved via SDS-PAGE and detected via western blotting using anti-B55α and GST antibodies. Densitometric quantitation was performed using ImageJ software.

### Immunoprecipitations

Whole-cell extracts (200–400 μg) were incubated with Myc- or Flag-conjugated beads (Sigma) for 3 hr or overnight at 4°C. Input samples were collected prior to antibody-conjugated bead incubation, and supernatants were taken post-incubation following sample centrifugation. Beads were then washed (5×) with complete DIP lysis buffer and proteins were eluted in 2× LSB. Proteins were resolved by SDS-PAGE and probed using antibodies against proteins of interest.

### PP2A/B55α purification

Stably expressing Flag-B55α 293T cells were expanded into six 15 cm tissue culture plates until they reached confluency. Cells were then harvested, washed 3× in cold 1× PBS, and then lysed in complete DIP lysis buffer for 1 hr. Lysates were then incubated with Flag-conjugated beads for 5 hr or overnight at 4°C. After washing the beads 8× with complete DIP lysis buffer, elution buffer containing 200 μg/mL DYKDDDDK peptide (GenScript) was added to samples and incubated with shaking 2× for 30 min each (eluates were collected after each incubation). Eluates containing purified PP2A/B55α complexes were combined with 25% glycerol and 1 mM DTT for –80°C storage.

### In vitro kinase and phosphatase assays

GST-tagged p107 constructs loaded on beads were phosphorylated with recombinant cyclin A/CDK2 (Invitrogen) in 200 nM ATP KAS buffer (5 mM HEPES pH 7.2, 1 mM MgCl$_2$, 0.5 mM MnCl$_2$, 0.1 mM DTT). We incubated samples with shaking at 37°C for 2 hr unless indicated. Reactions were stopped by adding 2× LSB and heated at 65°C when used for PAGE or immunoblots. In vitro phosphorylated substrates for phosphatase assays were washed 3× in complete DIP buffer, followed by addition of indicated concentrations of purified PP2A/B55α. Reactions were incubated with shaking at 37°C for times indicated, followed by addition of 2× LSB and boiling at 65°C for western blotting.

### Malachite Green Phosphatase Assay

The phosphatase activity of purified PP2A/B55α complexes was assessed using a commercial Ser/Thr phosphatase assay kit (EMD Millipore). Briefly, phosphatase-containing samples were incubated with synthetic p107-derived phosphopeptides for various time points at RT (up to 30 min). Malachite Green reagent with 0.01% Tween-20 was then added to quench the reaction. Absorbance was determined at 600 nm on a microplate reader.

### Cell cycle analysis

U-2 OS cells were co-transfected with the pBOS GFP-H2B plasmid (*Kanda et al., 1998*) along with the indicated plasmids using Lipofectamine 3000 in accordance with the manufacturer's recommendations

and allowed to express 48 hr prior to collection. Cells were fixed and stained with propidium iodide as previously described (*Kurimchak et al., 2013*). Cell cycle analysis was performed using the gated GFP-positive cells in FlowJo v10 software (BD Biosciences).

## Molecular modeling of a p107 peptide presented by the groove on B55α to the active site of PP2A/C

To dock R621 to B55α, we retrieved peptide structures with 'xHPRVx' from the PDB, then calculated dihedral angles ($\Phi$, $\varphi$, and $\omega$) of each residue of HPRV motif. The distance of each pair of peptide structures is the maximum distance of dihedral angles of residue pairs as previously described (*North et al., 2011*). The peptide structures were clustered by DBSCAN function in the R project (https://cran.r-project.org/web/packages/dbscan/index.html). We used the structure with best resolution (1G8M and chain A, resolution = 1.75 Å) in the largest cluster as our reference peptide. The peptide structure was moved to D197 of B55α in PDB:3DW8 by manual rotations and translations in PyMOL, such that the $NH_1/NH_2$ atoms of R621 were approximately 3 Å from $OD_1/OD_2$ atoms of D197. This model was subsequently refined with FlexPepDock web server (http://flexpepdock.furmanlab.cs.huji.ac.il/; *Figure 7A and B*).

To identify the likely phosphate position in the PP2A active site, we retrieved 68 structures from the PDB containing a phosphate group ($PO_4^{3-}$) and a pair of $Mn^{2+}$ ions from a search on the $Mn^{2+}$ ion on our website Protein Common Interface Database (ProtCID), http://dunbrack2.fccc.edu/ProtCiD/IPdbfam/PfamLigands.aspx?Ligand=MN (*Xu and Dunbrack, 2020*). These 68 structures contain 47 distinct UniProt proteins including four human phosphatase proteins (PP1A_HUMAN: PDB:4MOV, 4MOY, and 4MP0; PP1G_HUMAN: PDB:4UT2 and 4UT3; PPP5_HUMAN: PDB:1S95; and PR15A_HUMAN: PDB:4XPN). For each $Mn^{2+}$ ion, we calculated the distances from the four oxygen atom of the phosphate and saved the shortest distance. The average distance from $PO_4^{3-}$ to Mn ions is 2.38 Å and the standard deviation is 0.59 Å. We also examined a structure of PP5 (PDB:5HPE) that contains a substrate peptide (expressed as a tag separated by a linker from the C-terminus of PP5) with a Glu side chain as phosphomimetic. After aligning the proteins homologous to PP2A with bound phosphate, the structure of 5HPE with a Glu residue, and PP2A in PDB:3DW8, it is clear that the oxygen atoms of the phosphates and the Glu are in very similar positions relative to the $Mn^{2+}$ ions (*Figure 7—figure supplement 1*). We defined a structure of PP2A + B55α with the Glu phosphomimetic from PDB:5HPE and the HPRV peptide as 'the two-fragment model.'

To build p107 peptide (MPMEPLMHPRV), we searched the PDB and found 520 peptide structures that contain the sequence 'EPxxxPR,' which would connect phosphate (with E as a phosphomimetic of pSer) and Arg in the two binding sites. We calculated the distances between the $OE_1/OE_2$ of Glu and $NH_1/NH_2$ of Arg for these 520 structures to find peptides long enough to connect Glu and Arg residues. We superposed 217 peptide structures with a distance ≥20 Å onto the 'two-fragment model' by minimizing the distances of the $OE_1/OE_2$ and $NH_1/NH_2$ atoms of the peptide with those of the two-fragment model by *pair_fit* function in PyMOL (*Figure 7—figure supplement 1*, top), then calculated the sum of the displacements of the OE and NH atoms for each peptide from the equivalent atoms in the two-fragment model (*Figure 7—figure supplement 1*, middle left). The top 20 structures with minimum distances to the $OE_1/OE_2$ atoms of Glu and $NH_1/NH_2$ of Arg residues within the reference peptide fragments were selected and mutated into 'EPLMHPR,' and added 'MPM' residues and 'V' residue to build the p107 peptide 'MPMEPLMHPRV' in PyMOL (*Figure 7—figure supplement 1*, bottom). Each PP2A-B55α-p107 structure was refined with the FlexPepDock server and the top 10 models with the best Rosetta energy scores were saved (*Alford et al., 2017*). The model in which the $OE_1/OE_2$ atoms of Glu and $NH_1/NH_2$ atoms of Arg of p107 peptide has the minimum distances to the $Mn^{2+}$ ions and D197 was selected. To assemble our PP2A-p107 complex model, we then superposed 3DW8 onto this model to add the scaffold PP2A/A subunit (*Figure 7C and D*).

## NMR spectrometry

### Plasmid construction for NMR studies

B55α$_{M1-N447}$ gene was subcloned into a modified pCDNA3.4 vector (pcDNA3.4-K-GFP-RP1B). The pcDNA3.4-K-GFP-RP1B vector contains a Kozak consensus sequence, an N-terminal His6-green fluorescent protein-tag, a TEV cleavage site, and a multiple cloning site following the pcDNA3.4 human cytomegalovirus (CMV) promoter. This plasmid was amplified and purified using the NucleoBond

PC 500 Plasmid Maxiprep Kit (MACHEREY-NAGEL). P107$_{M612-S687}$ was subcloned into an MBP-fusion vector (pTHMT). The PTHMT vector contains an N-terminal His6-Maltose Binding Protein (MBP)-tag, a TEV cleavage site, and a multiple cloning site following the T7 promoter.

## Protein expression and purification for NMR studies

B55α$_{1-447}$ was expressed in Expi293F cells (Thermo Fisher) at a ratio of 1.0 µg DNA per mL of final transfection culture volume. Transfections were performed using 125 mL medium (Gibco Expi293 Expression Medium) in 250 mL baffled flasks (Corning) according to the manufacturer's protocol in a humidified incubator at 37°C and 8.0% CO$_2$ under shaking (125 rpm). On the day of transfection, the cell density was between 3–5 × 10$^6$ cells mL$^{-1}$. Prior to transfection, the Expi293F cells were seeded at 2.6 × 10$^6$ cells mL$^{-1}$ in 85% of the final transfection volume. B55α DNA was mixed with Opti-MEM Reduced Serum Medium (Thermo Fisher); in a separate tube, polyethylenimine (PEI) reagent was mixed with Opti-MEM medium. The DNA and PEI mixtures were combined and incubated for an additional 10 min. The final transfection mixture was then added to the cells. 2 mM (final concentration) valproic acid was added to the cells 18 hr post transfection. After an additional 24–28 hr, the cells were harvested and the pellet was stored at −80°C. P107$_{612-687}$ was expressed in *E. coli* BL21 DES cells (Agilent). Cells were grown in Luria Broth in the presence of the selective antibiotic at 37°C to an OD$_{600}$ of ~0.8, and expression was induced by addition of 1 mM IPTG. Induction proceeded for 5 hr at 37°C prior to harvesting by centrifugation at 6000 ×g for 15 min (Thermo Fisher). Cell pellets were stored at −80°C until purification. p107 cell pellets were resuspended in ice-cold lysis buffer (50 mM Tris pH 8.0, 0.5 M NaCl, 5 mM imidazole, 0.1% Triton X-100 containing a EDTA-free protease inhibitor tablet [Sigma]), lysed by high-pressure cell homogenization (Avestin C3 Emulsiflex), and centrifuged (35,000 ×g, 40 min, 4°C). The supernatant was loaded onto a HisTrap HP column (GE Healthcare) pre-equilibrated with Buffer A (50 mM Tris pH 8.0, 500 mM NaCl, and 5 mM imidazole) and was eluted using a linear gradient of Buffer B (50 mM Tris pH 8.0, 500 mM NaCl, 500 mM imidazole). Fractions containing the protein were pooled and dialyzed overnight at 4°C (50 mM Tris pH 7.5, 500 mM NaCl) with TEV protease to cleave the His$_6$-MBP-tag. The cleaved p107 was heated at 80°C for 10 min and centrifuged (29,000 ×g, 20 min, 4°C). The supernatant was concentrated to 3–4 mL and heated at 80°C for 10 min and centrifuged prior to SEC purification (SEC buffer: 20 mM sodium phosphate pH 6.5, 50 mM NaCl or 250 mM NaCl, 0.5 mM TCEP; column: Superdex 75 16/60, GE Healthcare).

B55α cell pellets were resuspended in ice-cold lysis buffer containing EDTA-free protease inhibitor tablet (Sigma) and rocked at 4°C for 20 min before centrifuged (42,000 ×g, 50 min, 4°C). The supernatant was mixed with GFP-nanobody-coupled agarose beads (AminoLink Plus Immobilization Kit, Thermo Fisher Scientific) and rocked at 4°C for 2 hr, and centrifuged (1000 ×g, 5 min, 4°C). The resin was washed twice with buffer (20 mM Tris, pH 7.5, 100 mM sodium chloride, 0.5 mM TCEP) by centrifugation. The His$_6$-GFP-tagged B55α-loaded resin was then suspended in 20 mL of the wash buffer and incubated with TEV protease overnight at 4°C. The cleaved product was centrifuged, and the supernatant was collected. The protein was further purified using anion exchange chromatography (Mono Q 5/50GL , GE Healthcare) pre-equilibrated with ion exchange buffer A (20 mM Tris, pH 7.5, 100 mM sodium chloride, 0.5 mM TCEP) and eluted using a linear gradient of Buffer B (20 mM Tris, pH 7.5, 1 M sodium chloride, 0.5 mM TCEP). Fractions containing B55α were identified using SDS-PAGE, pooled, and concentrated to designated concentration for experiments or stored at −80°C.

## NMR spectrometry data collection and processing

NMR data were recorded at 283 K on a Bruker Neo 600 MHz or 800 MHz ($^1$H Larmor frequency) NMR spectrometer equipped with a HCN TCI active z-gradient cryoprobe. NMR measurements of p107$_{612-687}$ were recorded using either $^{15}$N- or $^{15}$N,$^{13}$C-labeled protein at a final concentration of 6 or 200 µM in NMR buffer (20 mM sodium phosphate pH 6.8, 250 or 50 mM NaCl, 0.5 mM TCEP) and 90% H$_2$O/10% D$_2$O. Unlabeled B55α and $^{15}$N-labeled p107 was mixed at 1:1 ratio to form the complex. The sequence-specific backbone assignments of p107$_{612-687}$ were achieved using 3D triple resonance experiments including 2D [$^1$H,$^{15}$N] HSQC, 3D HNCA, 3D HN(CO)CA, 3D HN(CO)CACB, 3D HNCACB, 3D HNCO, and 3D HN(CA)CO. All NMR data were processed using Topspin 4.0.5 and analyzed using Cara. All NMR chemical shifts have been deposited in the BioMagResBank (BMRB: 28091).

## Biostatistics analysis

All experiments were performed in biological triplicate unless specified. Graphs depict calculated mean values with SD of triplicates. To assess significance, p-values were determined by Student's t-test and represented as follows: *<0.05, **<0.01, ***<0.001, and ****<0.0001.

## Acknowledgements

We thank Patrick Woodruff for technical assistance in early stages of this work. This work was supported in part by the National Institutes of Health Grants R01 GM117437 and R03 CA216134-01, a WW Smith charitable Trust Award, a FCCC/TU Nodal award (XG), R35 GM122517 (RLD) R01NS091336 and R01GM134683 (WP), and a Pre-Pilot Award from U54 CA221704 (ZZ and HF) and funding from NCI CCSG grant P30 CA006927 (XG and RLD).

## Additional information

### Funding

| Funder | Grant reference number | Author |
| --- | --- | --- |
| National Institute of General Medical Sciences | R01 GM117437 | Xavier Graña |
| National Cancer Institute | R03 CA216134-01 | Xavier Graña |
| WW Smith charitable Trust Award | no reference number | Xavier Graña |
| National Cancer Institute | P30 CA006927 | Xavier Graña Roland L Dunbrack |
| National Cancer Institute | U54 CA221704 | Holly Fowle Ziran Zhao |
| National Institute of General Medical Sciences | R01GM134683 | Wolfgang Peti |
| National Institute of Neurological Disorders and Stroke | R01NS091336 | Wolfgang Peti |

The funders had no role in study design, data collection and interpretation, or the decision to submit the work for publication.

### Author contributions

Holly Fowle, Brennan C McEwan, Xavier Graña, Conceptualization, Data curation, Formal analysis, Funding acquisition, Investigation, Methodology, Project administration, Resources, Supervision, Validation, Visualization, Writing – original draft, Writing – review and editing; Ziran Zhao, Conceptualization, Formal analysis, Funding acquisition, Investigation, Methodology, Writing – original draft; Qifang Xu, Conceptualization, Formal analysis, Funding acquisition, Investigation, Methodology, Software, Validation, Visualization, Writing – original draft; Jason S Wasserman, Xinru Wang, Formal analysis, Investigation, Methodology, Software, Validation, Visualization, Writing – original draft; Mary Adeyemi, Felicity Feiser, Alison N Kurimchak, Investigation, Methodology; Diba Atar, Methodology; Arminja N Kettenbach, Data curation, Formal analysis, Funding acquisition, Investigation, Methodology, Visualization, Writing – original draft, Writing – review and editing; Rebecca Page, Conceptualization, Investigation, Methodology, Supervision; Wolfgang Peti, Conceptualization, Formal analysis, Funding acquisition, Investigation, Methodology, Supervision, Writing – original draft, Writing – review and editing; Roland L Dunbrack, Conceptualization, Data curation, Formal analysis, Funding acquisition, Investigation, Methodology, Writing – original draft

### Author ORCIDs

Xinru Wang http://orcid.org/0000-0001-5994-707X
Arminja N Kettenbach http://orcid.org/0000-0003-3979-4576

Xavier Graña  http://orcid.org/0000-0001-7134-0473

**Decision letter and Author response**
Decision letter https://doi.org/10.7554/eLife.63181.sa1
Author response https://doi.org/10.7554/eLife.63181.sa2

## Additional files

### Supplementary files
• Transparent reporting form

### Data availability
Raw MS data for the data depicted in Figure 6B are available at MassIVE under accession number PXD028612. All NMR chemical shifts (Figure 1E-F) have been deposited in the BioMagResBank (BMRB: 28091). Source code for Figure 7 is a C# project, including retrieval of peptide structures from PDB and other sources such as PISCES, and calculation of distances and data analyses, available at GitHub (https://github.com/DunbrackLab/PP2A_PeptideDock; copy archived at swh:1:rev:24a52f-88800089b4a365df7837e240d5ea351bed). All other data generated or analysed during this study are included in the manuscript and supporting files. Source Data files have been provided.

The following datasets were generated:

| Author(s) | Year | Dataset title | Dataset URL | Database and Identifier |
| --- | --- | --- | --- | --- |
| Fowle H, Zhao Z, Xu Q, Wasserman JS, Wang X, Adeyemi M, Feiser F, Kurimchak A, Atar D, McEwan BC, Kettenbach AN, Page R, Peti W, Dunbrack RL, Graña X | 2021 | PP2A/B55a substrate recruitment as defined by the retinoblastoma-related protein p107 | https://doi.org/10.25345/C5426Q | MassIVE, 10.25345/C5426Q |
| Fowle H, Zhao Z, Xu Q, Wasserman JS, Wang X, Adeyemi M, Feiser F, Kurimchak A, Atar D, McEwan BC, Kettenbach AN, Page R, Peti W, Dunbrack RL, Graña X | 2021 | B55alpha-p107 spacer NMR chemical shifts | https://bmrb.io/data_library/summary/index.php?bmrbId=28091 | BMRBID, 28091 |

The following previously published dataset was used:

| Author(s) | Year | Dataset title | Dataset URL | Database and Identifier |
| --- | --- | --- | --- | --- |
| Xu Y, Chen Y, Zhang P, Jeffrey PD, Shi Y | 2008 | Structure of a Protein Phosphatase 2A Holoenzyme with B55 subunit | https://www.rcsb.org/structure/3DW8 | RCSB Protein Data Bank, 10.2210/pdb3DW8/pdb |

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
