## [Decision Letter]

**Acceptance summary:**

Phosphoprotein phosphatases (mainly PP1 and PP2A) make up the majority of serine/threonine phosphatase activity in the cell. While substrate recognition has been well studied for PP1 and PP2B, the substate recognition of PP2A holoenzymes are less understood. Recently, a substrate recognition motif LxxIxE was identified for B56/PP2A substrates. Here, Fowle et al., set to understand substrate recognition of B55/PP2A. Using a specific substrate of B55, p107, the authors identify a conserved binding motif (HxRVxxV) for recognition by B55 and show additional B55 substrates also contain this motif. The paper incorporates many complementary structural and biochemical assays to delineate the binding and recognition of substrates by B55.

**Decision letter after peer review:**

Thank you for submitting your article "PP2A/B55α substrate recruitment as defined by the retinoblastoma-related protein p107" for consideration by *eLife*. Your article has been reviewed by 3 peer reviewers, and the evaluation has been overseen by a Reviewing Editor and Philip Cole as the Senior Editor. The following individual involved in review of your submission has agreed to reveal their identity: Marcos Malumbres (Reviewer #3).

The reviewers have discussed the reviews with one another and the Reviewing Editor has drafted this decision to help you prepare a revised submission.

As the editors have judged that your manuscript is of interest, but as described below that additional experiments are required before it is published, we would like to draw your attention to changes in our revision policy that we have made in response to COVID-19 (https://elifesciences.org/articles/57162). First, because many researchers have temporarily lost access to the labs, we will give authors as much time as they need to submit revised manuscripts. We are also offering, if you choose, to post the manuscript to bioRxiv (if it is not already there) along with this decision letter and a formal designation that the manuscript is "in revision at eLife". Please let us know if you would like to pursue this option. (If your work is more suitable for medRxiv, you will need to post the preprint yourself, as the mechanisms for us to do so are still in development.)

Summary:

PP1 and PP2A make up the majority of serine/threonine phosphatase activity in the cell. While substrate recognition has been studied for PP1 and PP2B, the substate recognition of PP2A holoenzymes are less understood. Here, Fowle et al., set to understand substrate recognition of B55/PP2A. Using a specific substrate of B55α, p107, the authors identify a conserved binding motif (HxRVxxV) for recognition by B55α and show additional B55α substrates also contain this motif. This work incorporates many complementary structural and biochemical assays to delineate the binding and recognition of substrates by B55α.

Essential revisions:

1. What is the evidence that this motif is only recognized by B5alpha/PP2A and not other B55 family members? Are the residues identified in B5alpha critical to the p107 interaction (D197 and L225) conserved among all of the isoforms? If they are, can other B55 family members bind p107?

2. Have the authors looked for the HxRVxxV motif across the proteome? The author only state that they noticed that this motif was found in Tau, p130 and MAP2, but how many proteins contain these motifs? A list or understanding of the potential proteins which contain this motif could give researchers outside the field a link to understand the phosphatase important for their protein of interest.

3. For the P107 deletion mutants has the expression of each one been confirmed in Figure 1 and is decreased binding to PP2A B55alpha been normalized to the expression of these mutants.

4. Is the phosphorylation of p107 by CDK2 affecting the affinity of B55 binding to this substrate?

5. Have the authors considered measuring direct binding affinities using ITC/SPR for example to look at the effects of these various mutants in a cell free / in vitro system?

6. It would have been interesting to study the effects of the various B55 mutants on the endogenous phosphorylation of p107, Rb, and KSR?

7. To gain insights into the physiological role of the identified domain of p107 in PP2A-B55 binding and in the dephosphorylation of this protein, new "in cellulo" experiments using the full length p107 mutant protein have to be performed and its impact in the temporal pattern of dephosphorylation analyzed.

8. Figure 1D, it is obvious that in order to compare the levels of PP2A-B55 associated to each construct it is essential to normalize the levels of A and B55 signals to the quantity of protein that is recovered in each pulldown. As such, the levels of each GST construct in the pulldowns have to be measured by western blot and used to obtain the PP2A A our B55/GST-spacer ratio. Ratios can be then compared.

9. The authors state that: "a mutant lacking residues C-terminal of R2 binds B55α similarly to the full construct, indicating that residues C-terminal to the R2 domain are dispensable for B55α Binding”. Do "residues C-terminal of R2" mean full R2 region? If this is the case, this statement is not supported by Supplementary Figure 1B, where western blot of construct 2 and 7 display dramatically reduced B55 and A levels.

10. The authors tested the effect of KR residue mutation in the R1 and R2 regions in p107 dephosphorylation. KR mutants used for the R1 are R621A/K623A, the two mutants that were tested in Figure 1D and that were shown to impact B55 binding. However, they select K657A/R659A for R2 region. These two mutants were not tested in Figure 1D. Why do they introduce these mutants and not R647A that was investigated in Figure 1D? If the authors think that these residues are important, why did they not test them for its capacity to bind B55 in Figure 1D?

11. Other cdk-dependent phosphorylation sites on p107 that are essential for E2F binding have been described. Some of these sites are out of the spacer sequence. It will be interesting to know whether the dephosphorylation of these sites are dependent on PP2A-B55 and regulated by the mutants on the spacer sequence that decrease B55 binding.

12. Figure 4A and B. Dephosphorylation pattern of R1R2 control construct is drastically different in Figure 4A compared to 4B. In the first case, complete dephosphorylation does only take place upon two hours of incubation compared with fifteen minutes in the second. This is very weird if the same purified phosphatase is used in both experiments. In this line, I would expect a timing of few minutes for a total dephosphorylation when a purified phosphatase is used. Does it mean that phosphatase in Figure 4A lost activity?

13. "in vivo" experiments on the dephosphorylation of the non-binding p107 full length mutants have not been performed. To demonstrate that these residues are physiologically relevant for the physiological temporal p107 dephosphorylation pattern, these experiments must be done.

14. In the same line, to really show the involvement of the pST-x(5-10)-(RK)-Vxx(VI)R in Tau dephosphorylation by PP2A-B55 a direct mutant of this sequence of Tau should be checked.

15. What are the consequences of B55a-interaction mutants in p107 function? Is that mutant protein able to sustain cell cycle arrest?

16. Since the authors propose a new model/motif, it would be great to add some statistics on to what extent this motif is present in the numerous hits found in recent screens for B55 targets during mitotic exit. Is this motif present in B55 targets involved in non-cell-cycle (TAU) or cell-cycle targets? Is it equally present in proteins dephosphorylated during early versus late mitotic exit? Any hint into these questions may facilitate the impact of the model proposed in the biology of PP2A/B55.

---

## [Author Response]

*Essential revisions:*

1. What is the evidence that this motif is only recognized by B5alpha/PP2A and not other B55 family members? Are the residues identified in B5alpha critical to the p107 interaction (D197 and L225) conserved among all of the isoforms? If they are, can other B55 family members bind p107?

We thank the reviewer for bringing this to our attention. We have now clarified this in the manuscript (in the Results section, page 8, lines 13-15). We were not claiming that the interaction of p107 is exclusive to only B55a. Indeed, all residues in the top of the B55a β-propeller structure are highly conserved. These are the residues we experimentally verified (by mutation) to participate in the binding interaction (except residue E27, which lies in a protein segment not present in B55g). Therefore, most likely all members of the B55 family interact with substrates using this SLiM (same as what was shown for B56), as long as both B55 and the substrate are expressed in the same cellular location. Indeed, we have experimentally determined that p107 (using GST-R1 and GST-R1R2 constructs) pulls down B55d (not shown).

2. Have the authors looked for the HxRVxxV motif across the proteome? The author only state that they noticed that this motif was found in Tau, p130 and MAP2, but how many proteins contain these motifs? A list or understanding of the potential proteins which contain this motif could give researchers outside the field a link to understand the phosphatase important for their protein of interest.and a similar point made later: 14. Since the authors propose a new model/motif, it would be great to add some statistics on to what extent this motif is present in the numerous hits found in recent screens for B55 targets during mitotic exit. Is this motif present in B55 targets involved in non-cell-cycle (TAU) or cell-cycle targets? Is it equally present in proteins dephosphorylated during early versus late mitotic exit? Any hint into these questions may facilitate the impact of the model proposed in the biology of PP2A/B55.3. For the P107 deletion mutants has the expression of each one been confirmed in Figure 1 and is decreased binding to PP2A B55alpha been normalized to the expression of these mutants.and a similar point made later: 8. Figure 1D, it is obvious that in order to compare the levels of PP2A-B55 associated to each construct it is essential to normalize the levels of A and B55 signals to the quantity of protein that is recovered in each pulldown. As such, the levels of each GST construct in the pulldowns have to be measured by western blot and used to obtain the PP2A A our B55/GST-spacer ratio. Ratios can be then compared.

Mutant peptide competition and dephosphorylation analysis using p107 and validated with TAU led to our proposal of p[ST]-P-x(4,10)-[RK]-V-x-x-[VI]-R (see Figure 8A). Searching for this motif using ScanProsite yields 28 hits including p107, p130, TAU and MAP2. More degenerate versions of the SLiM (p[ST]-P-x(4,10)-[RK]-[VIL]x-x-[VILM]-[RK]) considering conserved amino acid substitutions identify 309 hits in 302 sequences, 1.5% of the proteins in the Swiss Human Proteome. However, as the last residue in the SLiM contributes minimally to binding p107:B55a binding despite is alignment with TAU (Figure 5C, 8A), we determined the frequency of the potential SLiM sequence (p[ST]-P-x(4,10)-[RK]-[VIL]-x-x-[VILM]-x), which retrieved 2831 hits in 2422 sequences (13.9% of the proteome) (Figure 8—figure supplement 1). However, considering that the SLiM would have to be conserved, in an intrinsically Disordered Region (IDR) and controlling dephosphorylation of a phosphosite(s), the frequency of a genuine SLiM is predictably much smaller. If our SLiM is functional in a significant fraction of B55a substrates, SLiM occurrence will be expected to relatively increase in datasets of B55a interactors or datasets where modulation of B55a expression results in phosphoproteome changes. This is in fact what we have seen and reported in a novel Figure 8—figure supplement 1. The fraction of proteins with the SLiM greatly increases among B55a interactors and potential B55a mitotic in vitro substrates (we have also provided an Excel table of the datasets, Figure 8—figure supplement 1-source data 1). This enrichment indicates that many substrates and B55 interactors potentially use this SLiM-based mechanism. Our follow up studies beyond the scope of this manuscript, have further validated the p[ST]-P-x(4,10)-[RK]-[VIL]-x-x-[VILM]-x SLiM by demonstrating that it is essential for binding/or dephosphorylation of two other B55a interactor/substrates including pRB.

The reviewer is correct, and we apologize for this oversight. We have substituted Supplementary Figure 1B, with a complete representative experiment that includes Coomassie Blue staining of the GST-p107 constructs. Quantitation of B55a signal was normalized by the levels of p107. The same was done for all replicates. This is now specified in the legend of Figure 1D (lines 17 to 19).

4. Is the phosphorylation of p107 by CDK2 affecting the affinity of B55 binding to this substrate?

This is a very interesting question and we have performed two independent experiments. First, we determined if mutation of GST-p107 R1R2 S615 to Glu or Ala affects binding. Author response image 1 shows that mutation to Glu, which mimics the phosphorylation charge, has little effect on binding.

**Author response image 1. sa2fig1:** Pull down assays with the indicated fusion proteins from HEK-293 lysates were performed and binding of B55_a_ was determined by western blot analysis. Two replicates of the experiment showed comparable results.

Using a separate approach, we determined if phosphorylated and unphosphorylated GST-R1R2 bind B55a comparably. Because B55a/PP2A dephosphorylate Cyclin A/CDK2 phosphorylated GST-R1R2 very efficiently, we performed the dephosphorylation step in the presence and in the absence of okadaic acid (OA), which potently inhibits PP2A activity. We noticed that OA diminished binding of GSTR1R2 to B55a, which suggests that when OA occupies the active site of PP2A/C it interferes with binding of p107 to B55a. However, phosphorylated p107 diƒod not bind better than dephosphorylated p107, indicating, as in the previous experiment, that phosphorylation of the target site does not increase the affinity of the substrate for the holoenzyme.

**Author response image 2. sa2fig2:** GST-R1R2 was phosphorylated with Cyclin A/CDK2 where indicated as described in the Manuscript. B55a binding to GST-R1R2 was determined in the presence or absence of OA to prevent dephosphorylation. Proteins were resolved via SDS-PAGE and detected by western blotting using anti-B55α and GST antibodies. Two replicates of the experiment showed comparable results.

5. Have the authors considered measuring direct binding affinities using ITC/SPR for example to look at the effects of these various mutants in a cell free / in vitro system?

The reviewer is correct using SPR to determine affinities using synthetic peptides to interrogate how substitution of individual residues with similar or differing properties affects affinity is an interesting approach. However, as it was recently shown by some of the authors of the manuscript (PMID: 31633908), these mutants often have unexpected effects. Especially, as they showed for the LxVP SLiM – a change in the LxVP SLiM sequence changed only the on-rate for binding and thus the binding K_D_ depends on the ensemble of confirmation the binding protein adopts (the specific sequence elements will restrict the phi/psi space). Thus, while we agree that this is an important line of investigation, due to the much more complex nature of this correlation this is beyond the scope of the work for this manuscript.

6. It would have been interesting to study the effects of the various B55 mutants on the endogenous phosphorylation of p107, Rb, and KSR?

Since antibodies specific to pRB phosphosites potentially targeted by B55a/PP2A are available, we have determined the effect of inducible expression of WT Flagtagged B55a and B55a_D197K_ on the phosphorylation of pRB in HEK-293 cells. Inducible expression of WT B55a, but not B55a_D197K_ results in time dependent dephosphorylation of endogenous pRB.

**Author response image 3. sa2fig3:** HEK-293 cells with doxicycline (Dox) inducible WT B55_a_ and B55_aD197K_ stable transgenes where treated with Dox for the indicated times and expression of indicated proteins and phospho pRB were determined by western blot analysis.

We have also determined the effect of upregulation of B55a in serum starved and restimulated BJ-TERT cells progressing through the cell cycle. We observe clear delays in pRB phosphorylation that correlate with delays in accumulation of cells in S phase.

**Author response image 4. sa2fig4:** BJ-TERT fibroblasts with doxicycline (Dox) inducible WT B55_a_ and B55_aD197K_ stable transgenes where grown to confluency treated with Dox for for 48 h, serum starved for 24 hours and then restimulated with Medium conaning 10% FBS for the indicated times. Expression of indicated proteins and phospho pRB was determined by western blot analysis.

7. To gain insights into the physiological role of the identified domain of p107 in PP2A-B55 binding and in the dephosphorylation of this protein, new "in cellulo" experiments using the full length p107 mutant protein have to be performed and its impact in the temporal pattern of dephosphorylation analyzed.

and a similar point made later: *13. "*in vivo*" experiments on the dephosphorylation of the non-binding p107 full length mutants have not been performed. To demonstrate that these residues are physiologically relevant for the physiological temporal p107 dephosphorylation pattern, these experiments must be done.*

We agree that these experiments are important and have generated a new fulllength p107 mutant with Ala substitution (p107_SLiM-MT_) of the essential SLiM residues (_619_HxRVxxV_625_ was substituted by _619_AxAAxxA_625_). As shown in new Figure 6D, the binding of WT B55a to p107_SLiM-MT_ is drastically diminished and this is accompanied by an expected significant increase in the phosphorylation of p107 S615 (this antibody only recognizes P-S615 in the p107 spacer and has no other consensus sites). Therefore, the p107 SLiM is essential to maintain steady state phosphorylation of at least p107 S615 and thus we can infer that B55a must be critical for the temporal regulation of this site.

8. Figure 1D, it is obvious that in order to compare the levels of PP2A-B55 associated to each construct it is essential to normalize the levels of A and B55 signals to the quantity of protein that is recovered in each pulldown. As such, the levels of each GST construct in the pulldowns have to be measured by western blot and used to obtain the PP2A A our B55/GST-spacer ratio. Ratios can be then compared.9. The authors state that: "a mutant lacking residues C-terminal of R2 binds B55α similarly to the full construct, indicating that residues C-terminal to the R2 domain are dispensable for B55α Binding”. Do "residues C-terminal of R2" mean full R2 region? If this is the case, this statement is not supported by Supplementary Figure 1B, where western blot of construct 2 and 7 display dramatically reduced B55 and A levels.

*"residues C-terminal of R2"* means any residues in the spacer after the end of conserved Region 2 (R2, after F_661_). R1 is required and necessary for binding, and any constructs lacking R1, do not bind B55a (Figure 1D top pulldown, lanes 6 and 7, bottom lane 5). Constructs, with R1 lacking R2, bind B55a, but constructs that contain both R1 and R2 show enhanced binding as R2 increases the avidity of p107 for B55a. As discussed in the Discussion section, “This R2 region includes a higher density of positively charged residues that enhance binding most likely via dynamic charged:charged interactions”.

10. The authors tested the effect of KR residue mutation in the R1 and R2 regions in p107 dephosphorylation. KR mutants used for the R1 are R621A/K623A, the two mutants that were tested in Figure 1D and that were shown to impact B55 binding. However, they select K657A/R659A for R2 region. These two mutants were not tested in Figure 1D. Why do they introduce these mutants and not R647A that was investigated in Figure 1D? If the authors think that these residues are important, why did they not test them for its capacity to bind B55 in Figure 1D?

In some of our earliest experiments we had compared binding of the GST-p107 spacer, to a previously made mutants that prevent cyclin E/A binding (KRRL-AAAA and RRL-AAA mutations to Ala). These mutations disrupt the RxL cyclin binding site. These experiments clearly showed that elimination of these residues reduce B55a binding, and confirmed reduced binding for cyclin A.

**Author response image 5. sa2fig5:** Pull down assays with the indicated fusion proteins from HEK-293 lysates were performed and binding of B55_a_ and Cyclin A was determined by western blot analysis. Two replicates of the experiment showed comparable results.

Since it was obvious that those residues were important for enhancing R2 avidity, we made the R1R2 K657A/R659A mutants when we started to measure dephosphorylation activity. We selected K657A/R659A to prevent disruption of the 558RxL motif required for Cyclin A binding. We have added clarificatory sentences to the text (Results page 2, lines 8 and 9; Discussion page 3, lines 3-4).

11. Other cdk-dependent phosphorylation sites on p107 that are essential for E2F binding have been described. Some of these sites are out of the spacer sequence. It will be interesting to know whether the dephosphorylation of these sites are dependent on PP2A-B55 and regulated by the mutants on the spacer sequence that decrease B55 binding.

We completely agree that this is interesting. We are studying other CDK sites on p107 that may be controlled by this SLiM or conserved SLiMs in other regions. We have started with S640, as we know gets dephosphorylated by B55a/PP2A (Figure 3C), but we do not know if its dephosphorylation is dependent on the SLiM, because we did not have a way to easily detect this phosphosite. We have now generated GSTp107R1R2 mutants that allow us to detect S640 independently of S615 with a CDK substrate antibody. We are generating R1 SLiM mutants to monitor S640 dependency. On the other hand, we have detected a potential degenerate SLiM in the C-terminus of p107, that is also conserved in pRB and could mediate dephosphorylation of C-terminal sites. Our preliminary data show binding of the pRB C-termini to B55a is dependent on the SLiM. This work is ongoing, but out of the scope of this manuscript.

12. Figure 4A and B. Dephosphorylation pattern of R1R2 control construct is drastically different in Figure 4A compared to 4B. In the first case, complete dephosphorylation does only take place upon two hours of incubation compared with fifteen minutes in the second. This is very weird if the same purified phosphatase is used in both experiments. In this line, I would expect a timing of few minutes for a total dephosphorylation when a purified phosphatase is used. Does it mean that phosphatase in Figure 4A lost activity?

We have used multiple batches of PP2A/B55a purified from human cells. While the purified holoenzyme is always active and suitable for experimentation the specific activity varies from batch to batch. Originally, we set up our assays to work in the range from 30 to 120 minutes, as we are using µg amounts of substrate, while both purified kinase and phosphatase holoenzymes are in the low nanogram range. When we started to introduce peptides for competition assays, we further optimized the experiment to be completed within one hour by increasing the amount of active phosphatase. As the reviewer suggests the relative amount of active holoenzyme used in 4A is lower than that used in 4B.

14. In the same line, to really show the involvement of the pST-x(5-10)-(RK)-Vxx(VI)R in Tau dephosphorylation by PP2A-B55 a direct mutant of this sequence of Tau should be checked.

As per this comment, we generated a full-length GST-FL-TAU mutant (P10636-8, 441 aa isoform). We found, GST-FL-TAU binds relatively more weakly than GSTp107-R1R2, despite the previous observation that it contains two lysine rich regions that may facilitate binding to B55a. We found that mutation of the SLiM did not disrupt binding in the context of the full-length TAU protein. As there are several charged:charged interactions that stabilize the interaction of the PP2A/B55a holoenzyme with FL-Tau, these may facilitate dephosphorylation of other sites in Tau and be sufficient to stabilize binding even in the absence of the SLiM. Our interpretation is that this TAU SLiM controls dephosphorylation of the Prolinedirected Thr_217_ amino terminal to the SLiM. In much studied phosphatases such as PP1 and PP2B, multiple SLiMs and charged-charged interaction contribute to overall binding.

15. What are the consequences of B55a-interaction mutants in p107 function? Is that mutant protein able to sustain cell cycle arrest?

Based on these questions, we felt that we needed to extend our findings depicted in Figure 6 with functional data. Therefore, we performed several co-transfection experiments in U-2 OS cells to determine the effect of B55a on p107 cell cycle effects and its interaction with E2F4. Our representative experiments are shown in Figure D-F. Together these show that B55a suppresses phosphorylation of at least S_615_ (Figure 6D), potently enhances p107 mediated cell cycle arrest (Figure 6E and F), but has no major effects on p107/E2F4 complex abundance (Figure 6—figure supplement 1). The data also show, that the p107 mediated G1 arrest is at least partially dependent on the p107 SLiM, and that mutation of the SLiM strongly diminishes the potent effects of B55a. We also found that cyclin E bypasses p107 cell cycle suppression without disrupting p107/E2F4 complexes. Altogether, suggests that B55a promotes p107 activation and cell cycle arrest at least in part by opposing cyclin E/CDK2 function likely by mechanisms independent of E2F4. Further work beyond the scope of this manuscript will be needed to fully address the mechanisms.